# Site-specific effects of neurosteroids on GABA$_A$ receptor activation and desensitization

Yusuke Sugasawa[1], Wayland WL Cheng[1], John R Bracamontes[1], Zi-Wei Chen[1,2], Lei Wang[1], Allison L Germann[1], Spencer R Pierce[1], Thomas C Senneff[1], Kathiresan Krishnan[3], David E Reichert[2,4], Douglas F Covey[1,2,3,5], Gustav Akk[1,2], Alex S Evers[1,2,3]*

[1]Department of Anesthesiology, Washington University in St. Louis, St. Louis, United States; [2]Taylor Family Institute for Innovative Psychiatric Research, Washington University in St. Louis, St. Louis, United States; [3]Department of Developmental Biology, Washington University in St. Louis, St. Louis, United States; [4]Department of Radiology, Washington University in St. Louis, St. Louis, United States; [5]Department of Psychiatry, Washington University in St. Louis, St. Louis, United States

**\*For correspondence:**
eversa@wustl.edu

**Competing interests:** The authors declare that no competing interests exist.

**Abstract** This study examines how site-specific binding to three identified neurosteroid-binding sites in the $\alpha_1\beta_3$ GABA$_A$ receptor (GABA$_A$R) contributes to neurosteroid allosteric modulation. We found that the potentiating neurosteroid, allopregnanolone, but not its inhibitory 3β-epimer epi-allopregnanolone, binds to the canonical $\beta_3$(+)–$\alpha_1$(-) intersubunit site that mediates receptor activation by neurosteroids. In contrast, both allopregnanolone and epi-allopregnanolone bind to intrasubunit sites in the $\beta_3$ subunit, promoting receptor desensitization and the $\alpha_1$ subunit promoting effects that vary between neurosteroids. Two neurosteroid analogues with diazirine moieties replacing the 3-hydroxyl (KK148 and KK150) bind to all three sites, but do not potentiate GABA$_A$R currents. KK148 is a desensitizing agent, whereas KK150 is devoid of allosteric activity. These compounds provide potential chemical scaffolds for neurosteroid antagonists. Collectively, these data show that differential occupancy and efficacy at three discrete neurosteroid-binding sites determine whether a neurosteroid has potentiating, inhibitory, or competitive antagonist activity on GABA$_A$Rs.

## Introduction

Neurosteroids (NS) are endogenous modulators of brain development and function and are important mediators of mood (*Belelli and Lambert, 2005*; *Mitchell et al., 2008*; *Represa and Ben-Ari, 2005*; *Grobin et al., 2006*). Exogenously administered NS analogues have been clinically used as anesthetics and anti-depressants and have therapeutic potential as anti-epileptics, neuroprotective agents and cognitive enhancers (*Belelli and Lambert, 2005*; *Mitchell et al., 2008*; *Akk et al., 2007*; *Reddy and Estes, 2016*; *Kharasch and Hollmann, 2015*; *Gunduz-Bruce et al., 2019*; *Zorumski et al., 2019*). The principal target of NS is the γ-aminobutyric acid type A receptor (GABA$_A$R). NS can either activate or inhibit GABA$_A$Rs. Positive allosteric modulatory NS (PAM-NS) such as allopregnanolone (3α5αP) potentiate the effect of GABA on GABA$_A$R currents at low concentrations and directly activate the receptors at higher concentrations (*Akk et al., 2007*; *Akk et al., 2010*; *Chen et al., 2019*; *Olsen, 2018*). Negative allosteric modulatory NS (NAM-NS), such as epi-allopregnanolone (3β5αP) or pregnenolone sulfate (PS) inhibit GABA$_A$R currents (*Akk et al., 2001*; *Wang et al., 2002*; *Shen et al., 2000*; *Lundgren et al., 2003*; *Seljeset et al., 2018*). In addition to

enhancing channel opening, PAM-NS increase the affinity of the GABA$_A$R for orthosteric ligand binding, an effect thought to be mechanistically linked to channel gating (*Chen et al., 2019*; *Harrison et al., 1987a*).

GABA$_A$Rs are pentameric ligand-gated ion channels (pLGIC) composed of two α-subunits (α$_{1-6}$), two β-subunits (β$_{1-3}$) and one additional subunit (γ$_{1-3}$, δ, ε, θ or π) (*Sigel and Steinmann, 2012*; *Sieghart, 2015*; *Olsen and Sieghart, 2008*). Each subunit is composed of a large extracellular domain (ECD), a transmembrane domain (TMD) formed by four membrane-spanning helices (TM1-4), a long intracellular loop between TM3 and TM4, and a short extracellular C-terminus (*Akk et al., 2007*; *Sigel and Steinmann, 2012*; *Sieghart, 2015*; *Laverty et al., 2019*). NS modulate GABA$_A$Rs by binding to sites within the TMDs (*Belelli and Lambert, 2005*; *Mitchell et al., 2008*; *Akk et al., 2007*; *Reddy and Estes, 2016*; *Chen et al., 2019*; *Miller et al., 2017*; *Laverty et al., 2017*; *Hosie et al., 2006*; *Hosie et al., 2009*; *Chen et al., 2018*; *Sugasawa et al., 2019*). Specifically, the α subunit TMDs are essential to the actions of PAM-NS (*Chen et al., 2019*; *Miller et al., 2017*; *Laverty et al., 2017*; *Chen et al., 2018*; *Sugasawa et al., 2019*). Mutagenesis studies in α$_1$β$_2$γ$_2$ GABA$_A$Rs have identified several residues in the α$_1$ subunit, notably Q242 and W246 in TM1, as critical to NS potentiation of GABA-elicited currents (*Hosie et al., 2006*; *Hosie et al., 2009*; *Akk et al., 2008*). Crystallographic studies have subsequently shown that, in homo-pentameric chimeric receptors in which the TMDs are derived from either α$_1$ (*Laverty et al., 2017*; *Chen et al., 2018*) or α$_5$ subunits (*Miller et al., 2017*), the NS 3α,21dihydroxy-5α-pregnan-20-one (3α5α-THDOC), pregnanolone and alphaxalone bind in a cleft between the α subunits, with the C3-hydroxyl substituent of the steroids interacting directly with Q242 in the α subunit (αQ242). PAM-NS activate these chimeric receptors, and their action is blocked by αQ242L and αQ242W mutations. These studies posit a single canonical intersubunit binding site for NS action that is conserved across the six α subunit isoforms (*Miller et al., 2017*; *Laverty et al., 2017*; *Chen et al., 2018*).

An alternative body of evidence suggests that PAM-NS modulation of GABA$_A$R function is mediated by multiple mechanisms and/or binding sites. Site-directed mutagenesis has identified multiple disparate residues on GABA$_A$Rs that affect NS-induced activation, suggestive of two NS-binding sites: one site mediating potentiation of GABA responses and the other mediating direct activation (*Hosie et al., 2006*; *Hosie et al., 2009*). Single channel electrophysiological studies (*Akk et al., 2007*; *Akk et al., 2010*; *Akk et al., 2004*) as well as studies examining neurosteroid modulation of [$^{35}$S]t-butylbicyclophosphorothionate (TBPS) binding (*Evers et al., 2010*), have also identified multiple distinct effects of NS, with various structural analogues producing some or all of these effects, consistent with multiple NS-binding sites (*Hosie et al., 2006*; *Hosie et al., 2009*). Our recent photo-labeling studies have confirmed that there are multiple PAM-NS-binding sites on α$_1$β$_3$ GABA$_A$Rs (*Chen et al., 2019*). In addition to the canonical site at the interface between the TMDs of adjacent subunits (intersubunit site) (*Chen et al., 2019*; *Miller et al., 2017*; *Laverty et al., 2017*; *Chen et al., 2018*), we identified NS-binding sites within the α-helical bundles of both the α$_1$ and β$_3$ subunits (intrasubunit sites) of α$_1$β$_3$ GABA$_A$Rs (*Chen et al., 2019*). 3α5αP binds to all three sites (*Chen et al., 2019*); mutagenesis of these sites suggests that the intersubunit and α$_1$ intrasubunit sites, but not the β$_3$ intrasubunit site, contribute to 3α5αP PAM activity (*Chen et al., 2019*). A functional effect for NS binding to the β$_3$ intrasubunit site has not been identified.

The 3α-hydroxyl (3α-OH) group is critical to NS activation of GABA$_A$Rs and 3β-OH NS lack PAM activity (*Akk et al., 2007*; *Wang et al., 2002*). Indeed, many 3β-OH NS are GABA$_A$R NAMs (*Wang et al., 2002*; *Lundgren et al., 2003*). While molecular docking studies have suggested that the 3β-OH NS epi-pregnanolone (3β5βP) should compete for binding with PAM-NS (*Miller et al., 2017*), 3β-OH NS are non-competitive inhibitors with respect to GABA and 3α-OH NS, indicating that they are unlikely to act at the canonical PAM-binding site (*Akk et al., 2007*; *Wang et al., 2002*). Steroids with a sulfate rather than a hydroxyl at the 3-carbon are also GABA$_A$R NAMs thought to act at sites distinct from GABA$_A$R PAMs (*Akk et al., 2007*; *Akk et al., 2001*; *Wang et al., 2002*; *Seljeset et al., 2018*; *Park-Chung et al., 1999*). The precise location of this site is unclear, but crystallographic studies have demonstrated a possible binding site between TM3 and TM4 on the intracellular end of the α-subunit TMD (*Seljeset et al., 2018*; *Laverty et al., 2017*). While 3β-OH NS and PS both inhibit GABA$_A$Rs, they likely act via interactions with distinct sites (*Akk et al., 2007*; *Akk et al., 2001*; *Wang et al., 2002*; *Lundgren et al., 2003*; *Seljeset et al., 2018*; *Miller et al., 2017*; *Laverty et al., 2017*).

The goal of the current study was to determine the specific sites underlying the PAM and NAM actions of NS. We hypothesized that various NS analogues preferentially bind to one or more of the three NS-binding sites in the $\alpha_1\beta_3$ GABA$_A$R, stabilizing distinct conformational states (i.e. resting, open or desensitized). To achieve this goal, we used two endogenous NS, the PAM-NS 3$\alpha$5$\alpha$P and the NAM-NS 3$\beta$5$\alpha$P and two NS analogues, KK148 and KK150, in which a diazirine replaced the function-critical 3-OH group (*Jiang et al., 2016*). We examined site-specific NS binding and effects using NS photolabeling (*Sugasawa et al., 2019*; *Budelier et al., 2017*; *Budelier et al., 2019*; *Cheng et al., 2018*) and measurements of channel gating and orthosteric ligand binding. The NS lacking a 3$\alpha$-OH were devoid of PAM-NS activity, but surprisingly, KK148 and 3$\beta$5$\alpha$P enhanced the affinity of [$^3$H]muscimol binding. We interpret this finding as evidence that these compounds preferentially bind to and stabilize desensitized receptors, since both open and desensitized GABA$_A$R exhibit enhanced orthosteric ligand-binding affinity (*Chang et al., 2002*).

The results show that 3$\alpha$5$\alpha$P binds to the canonical $\beta$(+)–$\alpha$(-) intersubunit site, stabilizing the open state of the receptor, whereas the 3-diazirinyl NS (KK148 and KK150) bind to this site but do not promote channel opening, and 3$\beta$5$\alpha$P does not occupy this site. These data indicate that NS binding to the intersubunit sites is largely responsible for PAM activity and that the 3$\alpha$-OH is critical for NS activation. In contrast, 3$\alpha$5$\alpha$P, 3$\beta$5$\alpha$P and the 3-diazirinyl NS all bind to both the $\alpha_1$ and $\beta_3$ intrasubunit sites. Occupancy of the intrasubunit sites by 3$\alpha$5$\alpha$P, 3$\beta$5$\alpha$P and KK148 promotes receptor desensitization. KK150 occupies all three NS-binding sites on $\alpha_1\beta_3$ GABA$_A$Rs, but produces minimal functional effect suggesting a possible scaffold for a general NS antagonist. These results shed new light on the mechanisms of NS allosteric modulation of channel function, and demonstrate a novel pharmacology in which related ligands bind to different subsets of functional sites on the same protein, each in a state-dependent manner, with the actions at these sites summating to produce a net physiological effect.

## Results

### Distinct patterns of NS potentiation and enhancement of muscimol binding

The endogenous NS, 3$\alpha$5$\alpha$P is known to potentiate GABA-elicited currents (*Figure 1A*) and enhance [$^3$H]muscimol binding to $\alpha_1\beta_3$ GABA$_A$Rs (*Figure 1E*; *Chen et al., 2019*; *Harrison et al., 1987a*). We examined a series of NS analogues with different stereochemistries or substituents in the 3- and 17-positions: 3$\beta$5$\alpha$P, KK148, and KK150 (structures shown in *Figure 1B–D*) for their ability to potentiate GABA-elicited currents and enhance orthosteric agonist ([$^3$H]muscimol) binding. 3$\beta$5$\alpha$P is the 3$\beta$-epimer of 3$\alpha$5$\alpha$P. KK148 and KK150 are NS analogue photolabeling reagents, which have a 3-diazirinyl moiety instead of the 3-OH, and differ from each other by the stereochemistry of the 17-ether linkage (*Jiang et al., 2016*). We observed a discrepancy between the ability of these compounds to potentiate GABA-elicited currents and their ability to enhance [$^3$H]muscimol binding in $\alpha_1\beta_3$ GABA$_A$Rs. None of the NS analogues lacking a 3$\alpha$-OH potentiated GABA-elicited currents (*Figure 1B–D*). However, both 3$\beta$5$\alpha$P and KK148 significantly enhanced [$^3$H]muscimol binding (*Figure 1E*). KK150, in contrast, did not potentiate GABA-elicited currents and minimally enhanced [$^3$H]muscimol binding (*Figure 1D–E*). Collectively, these data show that, NS analogues with different stereochemistry or substituents at the 3- and 17-positions show distinct patterns in modulation of $\alpha_1\beta_3$ GABA$_A$R currents and orthosteric ligand binding. We hypothesized that these patterns are a consequence of the various NS analogues stabilizing distinct conformational states of the GABA$_A$R, possibly by binding and acting at different sites. Notably, the compounds with a 3-OH (3$\alpha$5$\alpha$P, 3$\beta$5$\alpha$P) are 10-fold more potent than those with a 3-diazirine (KK148, KK150) in enhancing [$^3$H]muscimol binding (*Figure 1E*), suggesting that the 3-OH is an important determinant of binding affinity to the site(s) mediating these effects.

### State-specific actions of NS analogues

To determine why 3$\beta$5$\alpha$P and KK148 enhance [$^3$H]muscimol binding but do not potentiate $\alpha_1\beta_3$ GABA$_A$R currents, we first considered the possibility that 3$\beta$5$\alpha$P- and KK148-induced enhancement of [$^3$H]muscimol binding is a selective effect on intracellular GABA$_A$Rs, since the radioligand binding assay was performed on total membrane homogenates, whereas the electrophysiological assays

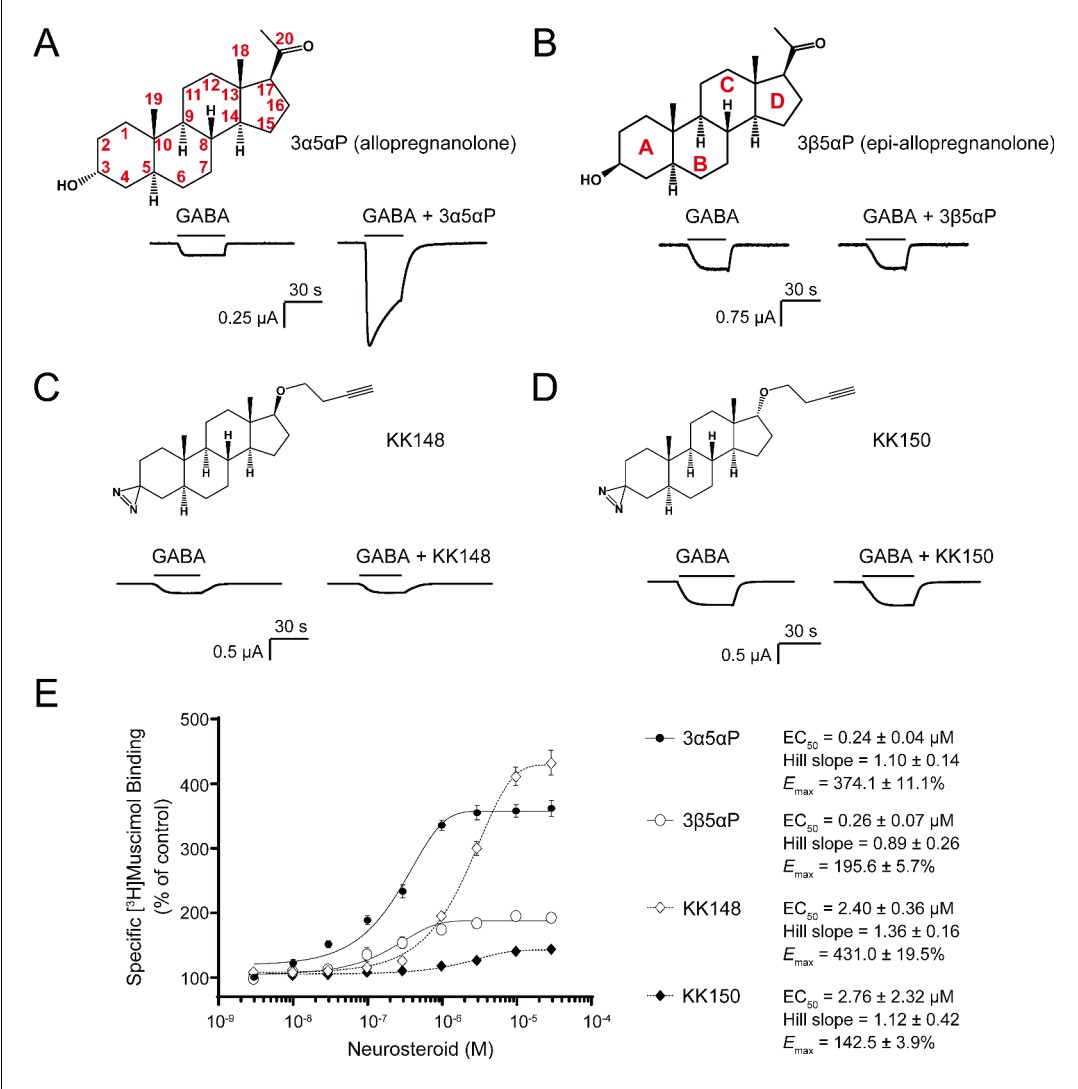

**Figure 1.** Distinct neurosteroid effects on potentiation of GABA$_A$R currents and modulation of [³H]muscimol binding. (**A**) Structure of allopregnanolone (3α5αP) with carbon atoms numbered and sample current traces from α$_1$β$_3$ GABA$_A$R activated by 0.3 μM GABA showing potentiation by 10 μM 3α5αP. The traces were recorded from the same cell. (**B**), (**C**) and (**D**) Structures of epi-allopregnanolone (3β5αP) with steroid rings labeled, neurosteroid analogue photolabeling reagents KK148 and KK150, respectively, and sample current traces from α$_1$β$_3$ GABA$_A$R activated by 0.3 μM GABA showing the absence of potentiation by 10 μM neurosteroids. Each pair of traces was recorded from the same cell. (**E**) Concentration-response relationship for neurosteroid modulation of [³H]muscimol binding to α$_1$β$_3$ GABA$_A$R. 3 nM–30 μM neurosteroids modulate [³H]muscimol (3 nM) binding in a concentration-dependent manner. Data points, EC$_{50}$, Hill slope and maximal effect value [$E_{max}$ (% of control): 100% means no effect] are presented as mean ± SEM ($n$ = 6 for 3α5αP and KK148; $n$ = 3 for 3β5αP and KK150).

The online version of this article includes the following figure supplement(s) for figure 1:

**Figure supplement 1.** Neurosteroid modulation of muscimol binding to intact cells.

report only from cell surface channels. NS are known to have effects on intracellular GABA$_A$Rs and have been shown to accelerate GABA$_A$R trafficking (*Abramian et al., 2014*; *Comenencia-Ortiz et al., 2014*; *Smith et al., 2007*). To test this possibility, we examined [³H]muscimol binding in intact cells (i.e. binding to receptors only in the plasma membrane) (*Vauquelin et al., 2015*; *Bylund et al., 2004*; *Bylund and Toews, 1993*) compared to permeabilized cells (plasma membranes plus intracellular membranes). Notably, [³H]muscimol binding was twofold greater in permeabilized cells than in intact cells, indicating a significant population of intracellular GABA$_A$Rs. KK148 enhanced [³H]muscimol binding in intact cells as much or more than in permeabilized cells, indicating

that this effect is not a result of selective NS actions on intracellular receptors (*Figure 1—figure supplement 1*).

A second possibility is that 3β5αP and KK148 selectively bind to and stabilize a high-affinity non-conducting state, such as a pre-active (*Gielen and Corringer, 2018*) or a desensitized conformation of the $GABA_AR$. This is expected to result in inhibition of receptor function; however, the magnitude of the effect may be small under the experimental conditions used to generate the traces in *Figure 1*. To examine the inhibitory effect of these NS analogues, we activated $\alpha_1\beta_3$ $GABA_AR$ with a saturating concentration (1 mM) of GABA and tested the effect of the NS on steady-state currents (*Germann et al., 2019a*). KK148 and 3β5αP both decreased steady-state currents (*Figure 2A and C*), whereas KK150 did not (*Figure 2B*). To further delineate the electrophysiological effects of these compounds, we focused on 3β5αP, since it is an endogenous NS and we had limited availability of KK148. Co-application of 3β5αP with 1 mM GABA preferentially inhibited steady-state rather than peak currents (*Figure 2—figure supplement 1*). While this result is consistent with stabilization of a desensitized state rather than a pre-active state, it is ambiguous because it is possible that the steroid has a slower onset than GABA, thus minimizing the effect on peak current. Additional evidence that 3β-NAM-NS stabilize a desensitized state includes studies examining their effects on inhibitory post-synaptic currents (*Wang et al., 2002*) and single channel currents (*Akk et al., 2001*). The evidence that NAM-NS stabilize a desensitized rather than a pre-active state is more thoroughly explored in the Discussion. In the ensuing text, we refer to the inhibition of steady-state current as desensitization.

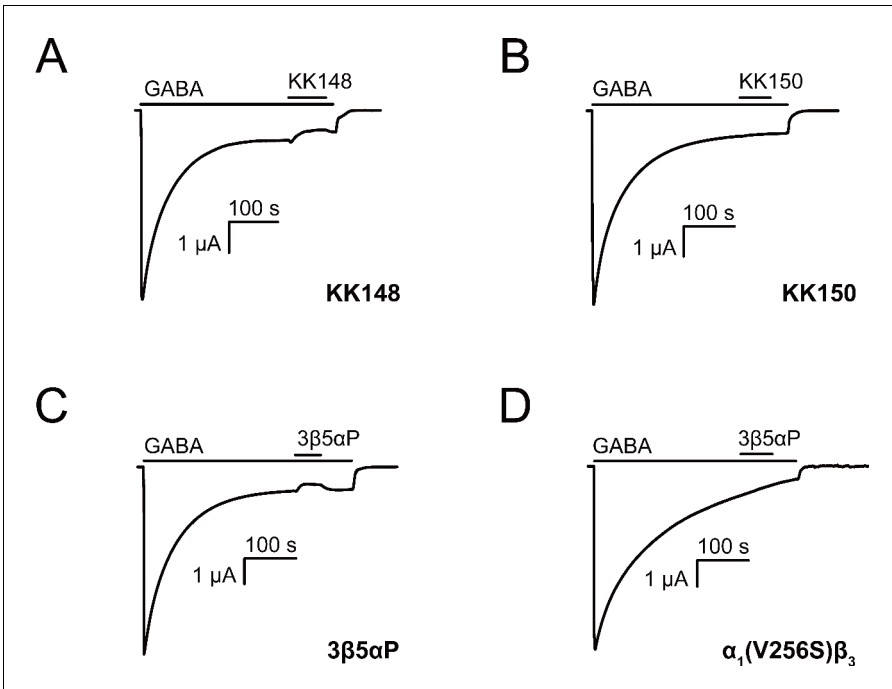

**Figure 2.** Neurosteroids promote steady-state desensitization of $\alpha_1\beta_3$ $GABA_ARs$. Representative traces showing the effects of KK148, KK150 and epi-allopregnanolone (3β5αP) on maximal steady-state GABA-elicited currents. $\alpha_1\beta_3$ $GABA_ARs$ expressed in *Xenopus laevis* oocytes were activated with 1 mM GABA to maximally activate $GABA_AR$ current. (**A–C**) The effect of KK148 (10 μM), KK150 (10 μM) and 3β5αP (3 μM) on steady-state current. (**D**) The effect of 3β5αP (3 μM) on steady-state current in $\alpha_1\beta_3$ $GABA_ARs$ containing the $\alpha_1$V256S mutation, known to eliminate NS-induced desensitization. The results show that 3β5αP and KK148 reduce steady-state currents, consistent with enhanced desensitization, whereas KK150 does not. The effect of 3β5αP on steady-state currents is eliminated by the $\alpha_1$V256S mutation, consistent with 3β5αP enhancing desensitization rather than producing channel block.

The online version of this article includes the following figure supplement(s) for figure 2:

**Figure supplement 1.** Co-application of epi-allopregnanolone with a saturating concentration of GABA.

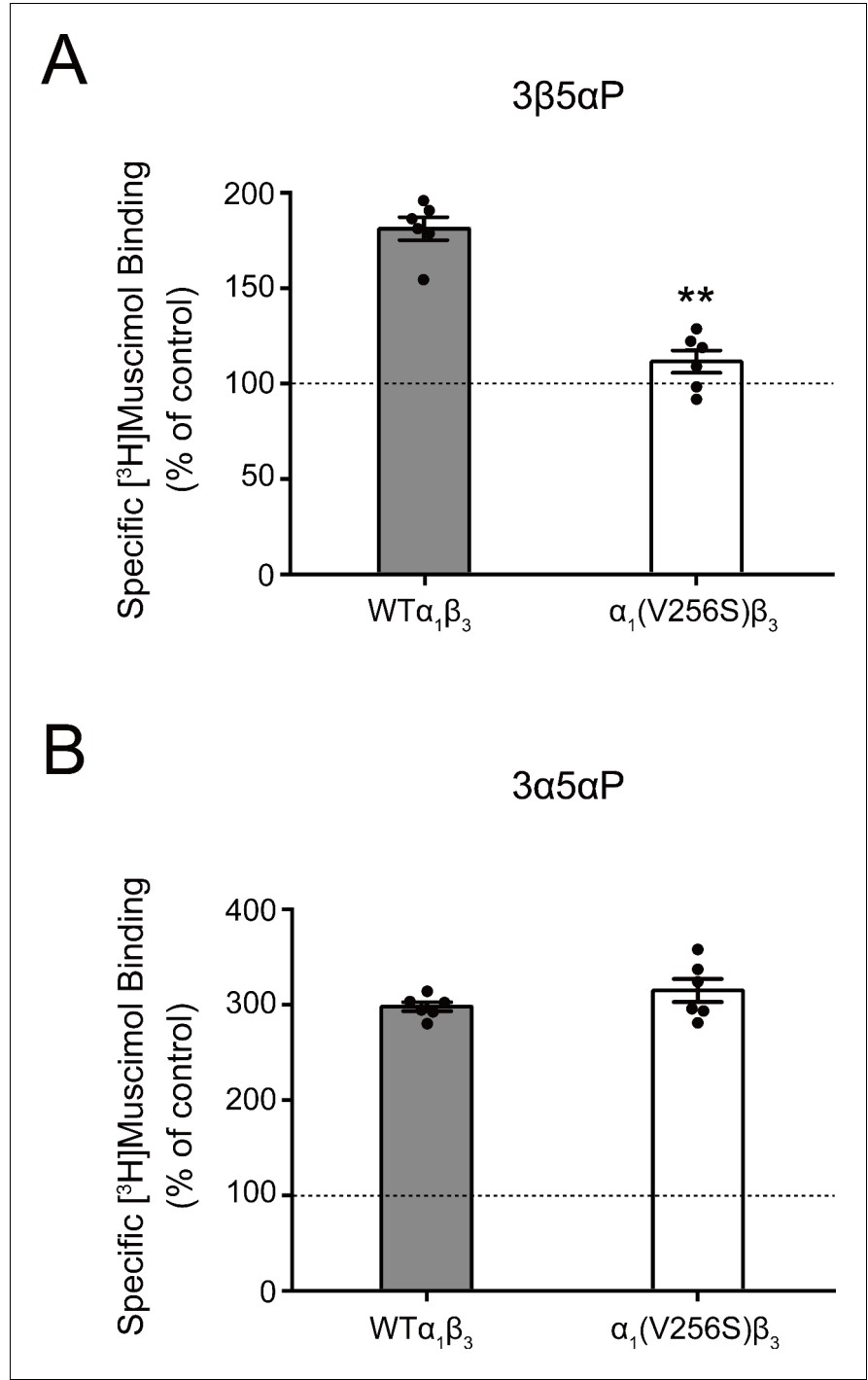

**Figure 3.** Effect of $\alpha_1$(V256S)$\beta_3$ mutation on neurosteroid enhancement of [3H]muscimol binding. (**A**) Enhancement of specific [3H]muscimol (3 nM) binding to $\alpha_1\beta_3$ GABA$_A$R WT by 10 μM epi-allopregnanolone (3$\beta$5$\alpha$P) is absent in $\alpha_1$(V256S)$\beta_3$ GABA$_A$R. (**B**) Enhancement of [3H]muscimol binding by 10 μM allopregnanolone (3$\alpha$5$\alpha$P) is unaffected by the $\alpha_1$V256S mutation. These data indicate that 3$\beta$5$\alpha$P enhancement of orthosteric ligand binding requires receptor desensitization, whereas 3$\alpha$5$\alpha$P does not. Statistical differences are compared using unpaired *t*-test ($n$ = 6,± SEM). **p<0.01 vs. WT.

The inhibitory effect of 3$\beta$5$\alpha$P was not observed in receptors with the $\alpha_1$(V256S) TM2 pore-lining mutation, which was previously shown to remove the inhibitory effects of sulfated steroids (*Akk et al., 2001*; *Wang et al., 2002*; *Figure 2D*). Although both 3$\alpha$5$\alpha$P and 3$\beta$5$\alpha$P enhance [3H]

muscimol binding, the former predominantly results in receptor activation, whereas the latter results in inhibition. Consistent with this, the $\alpha_1$(V256S)$\beta_3$ mutation which abolishes NS-induced inhibition (*Akk et al., 2001*; *Wang et al., 2002*) eliminated [$^3$H]muscimol binding enhancement by 3$\beta$5$\alpha$P but not 3$\alpha$5$\alpha$P (*Figure 3*). We infer that 3$\alpha$5$\alpha$P increases [$^3$H]muscimol binding by stabilizing an active state of the receptor. In contrast, 3$\beta$5$\alpha$P increases [$^3$H]muscimol binding by stabilizing a desensitized state of the receptor; this effect is eliminated in the $\alpha_1$(V256S)$\beta_3$ receptor. The mechanisms of enhancement of [$^3$H]muscimol binding by allosteric activators and inhibitors are described in detail in our recent publication (*Akk et al., 2020*). Collectively, these data indicate that 3$\beta$5$\alpha$P and KK148 enhance orthosteric ligand affinity by stabilizing a desensitized state of the GABA$_A$R.

## Quantitative comparison of the effects of 3$\beta$5$\alpha$P on [$^3$H]muscimol binding and receptor desensitization

While there is qualitative agreement between the relative effects of the various NS analogues on orthosteric ligand binding and receptor desensitization, there is a quantitative discrepancy in the magnitude of the effects. For example, 3$\beta$5$\alpha$P enhances [$^3$H]muscimol binding by two-fold (*Figure 1E*), whereas it reduces steady-state current by only ~25% (*Figure 2C*). To address this difference, we considered that the radioligand binding and electrophysiological assays are performed under different experimental conditions. The radioligand-binding studies are performed using low [$^3$H]muscimol concentrations to allow for sufficient dynamic range of ligand binding. In contrast, the desensitization experiments are performed at high orthosteric ligand (GABA) concentration to achieve high peak open probability and steady-state receptor desensitization, thus minimizing the number of channels in the resting state. To address the quantitative differences in results from the two assays, we analyzed the electrophysiological data in the framework of the three-state Resting-Open-Desensitized model (*Germann et al., 2019a*; *Germann et al., 2019b*). We assumed that both the open and desensitized states had higher affinity for muscimol than the resting state, and that the affinities were similar and could be treated as equal. We then calculated the predicted occupancy of the high-affinity states (P$_{open}$ + P$_{desensitized}$) using parameters derived from the functional responses, to compare to the observed changes in binding. The raw current amplitudes of peak and steady-state responses were converted to units of open probability as described previously in detail (*Eaton et al., 2016*), and the probabilities of being in the open (P$_{open}$) or desensitized (P$_{desensitized}$) states were calculated for different experimental conditions (see Materials and methods).

Application of 1 mM GABA elicited a current response that had a peak P$_{open}$ of 0.71 ± 0.25 (mean ± SD; $n$ = 16). The P$_{open}$ of the steady-state response was 0.121 ± 0.033 ($n$ = 7), that was reduced to 0.077 ± 0.013 ($n$ = 5) with 3 $\mu$M 3$\beta$5$\alpha$P. Analysis of steady-state currents using the Resting-Open-Desensitized model indicates that the steady-state P$_{desensitized}$ is 0.829 in the presence of GABA, and 0.892 in the presence of GABA + steroid. The relatively small increase in the sum of (P$_{open}$ + P$_{desensitized}$) (from 0.95 to 0.97) is due to the use of saturating GABA in these experiments.

To compare the data from the radioligand binding and electrophysiology experiments, we exposed oocytes containing $\alpha_1\beta_3$ GABA$_A$Rs to 20 nM muscimol and recorded currents before and after co-application of 3 $\mu$M 3$\beta$5$\alpha$P. The percent reduction in steady-state current following 3$\beta$5$\alpha$P exposure was measured and used to estimate the relative probabilities of resting, open and desensitized receptors. The application of 20 nM muscimol elicited a peak response with P$_{open}$ of 0.012 ± 0.004 ($n$ = 6). The steady-state P$_{open}$ was 0.011 ± 0.004. In the same cells, subsequent exposure to 3 $\mu$M 3$\beta$5$\alpha$P reduced the steady-state P$_{open}$ to 0.009 ± 0.004 (p=0.0174; paired $t$-test). The calculated steady-state P$_{desensitized}$ was 0.1001 in the presence of muscimol, and 0.2168 in the presence of muscimol + 3$\beta$5$\alpha$P. Thus, there is a predicted two-fold increase in the sum of (P$_{open}$ + P$_{desensitized}$) when the steroid is combined with muscimol, consistent with the doubling of muscimol binding caused by 3$\beta$5$\alpha$P in the [$^3$H]muscimol binding experiments (*Figure 1E*). While the measured changes in current are small, they are precise because each experiment served as its own control; a steady-state current was achieved during continuous agonist administration and the response to 3$\beta$5$\alpha$P was then measured. Overall, these data indicate that when P$_{resting}$ is high (low orthosteric ligand concentration), an agent that stabilizes desensitized receptors may produce a small decrease in steady-state current, but a relatively large increase in the occupancy of desensitized state, at the expense of resting receptors. Conversely, with high orthosteric ligand concentrations (low P$_{resting}$), a desensitizing ligand produces a relatively larger change in steady-state current as open receptors

are converted to desensitized receptors with minimal effect on the sum occupancy of high-affinity states.

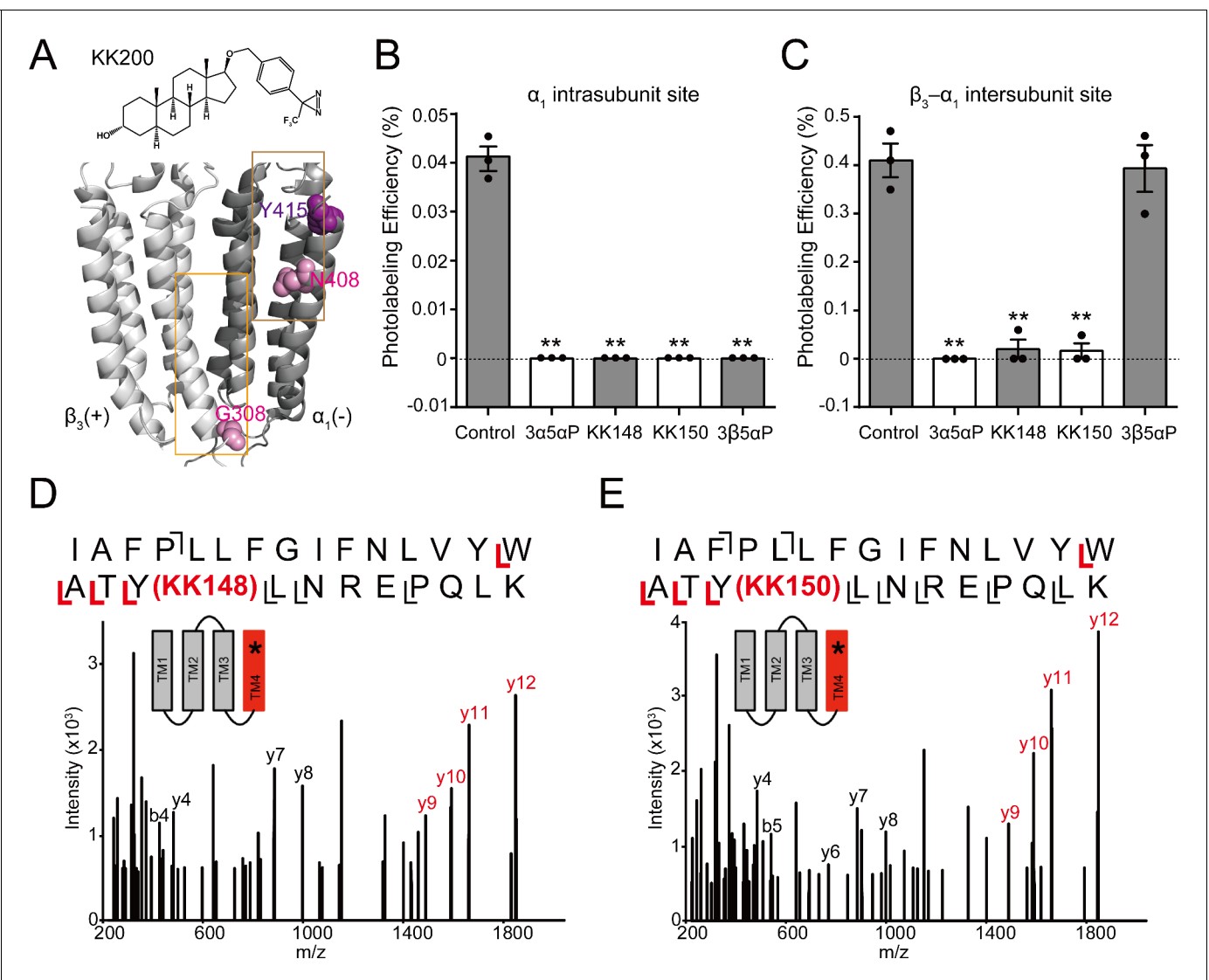

**Figure 4.** Competitive prevention of neurosteroid photolabeling at an intersubunit and intrasubunit site. (**A**) Structures of the neurosteroid photolabeling reagent KK200 and the $\alpha_1\beta_3$ GABA$_A$R-TMDs highlighting the residues G308 in the $\beta_3$(+)–$\alpha_1$(-) intersubunit site and N408 in the $\alpha_1$ intrasubunit site previously identified by KK200 photolabeling in pink. Shown in purple is Y415 in the $\alpha_1$ intrasubunit site, which is photolabeled by KK148 and KK150. Adjacent $\beta_3$(+) and $\alpha_1$(-) subunits are shown and the channel pore is behind the subunits. (**B**) Photolabeling efficiency of $\alpha_1$ subunit TM4 ($\alpha_1$ intrasubunit site) in $\alpha_1\beta_3$ GABA$_A$R by 3 µM KK200 in the absence or presence of 30 µM allopregnanolone (3$\alpha$5$\alpha$P), KK148, KK150, and epi-allopregnanolone (3$\beta$5$\alpha$P). Statistical differences are analyzed using one-way ANOVA with Bonferroni's multiple comparisons test ($n = 3, \pm$ SEM). \*\*$p<0.01$ vs. control. (**C**) Same as (**B**) for $\beta_3$ subunit TM3 [$\beta_3$(+)–$\alpha_1$(-) intersubunit site, $n = 3, \pm$ SEM]. (**D**) HCD fragmentation spectrum of the $\alpha_1$ subunit TM4 tryptic peptide photolabeled by 30 µM KK148. Red and black indicate fragment ions that do or do not contain KK148, respectively. The schematic highlight in red identifies the TMD being analyzed and the asterisk denotes the approximate location of KK148. (**E**) Same as (**D**) photolabeled by 30 µM KK150.

The online version of this article includes the following figure supplement(s) for figure 4:

**Figure supplement 1.** Extracted ion chromatograms of labeled and unlabeled $\beta_3$ subunit TM4 peptides.

**Figure supplement 2.** Fragmentation spectrum of unlabeled $\alpha_1$ subunit TM4 peptide.

## Binding site selectivity for NS analogues

To determine whether KK148 and 3β5αP stabilize a desensitized conformation of the GABA$_A$R by selectively binding to one or more of the identified NS-binding sites on the GABA$_A$R (*Chen et al., 2019*), we first determined which of the identified NS sites they bind. We have previously shown that the 3α5αP-analogue photolabeling reagent, KK200 labels the β$_3$(+)–α$_1$(-) intersubunit (β$_3$G308) and α$_1$ intrasubunit (α$_1$N408) sites on α$_1$β$_3$ GABA$_A$Rs (*Figure 4A*), and that photolabeling can be prevented by a 10-fold excess of 3α5αP (*Chen et al., 2019*). As a first step to determine the binding sites for 3β5αP, KK148 or KK150, we examined whether a 10-fold excess of these compounds (30 μM) prevented KK200 (3 μM) photolabeling of either binding site. Photolabeling was performed on membranes from HEK293 cells transfected with epitope-tagged α$_{1His-FLAG}$β$_3$ receptors, mimicking

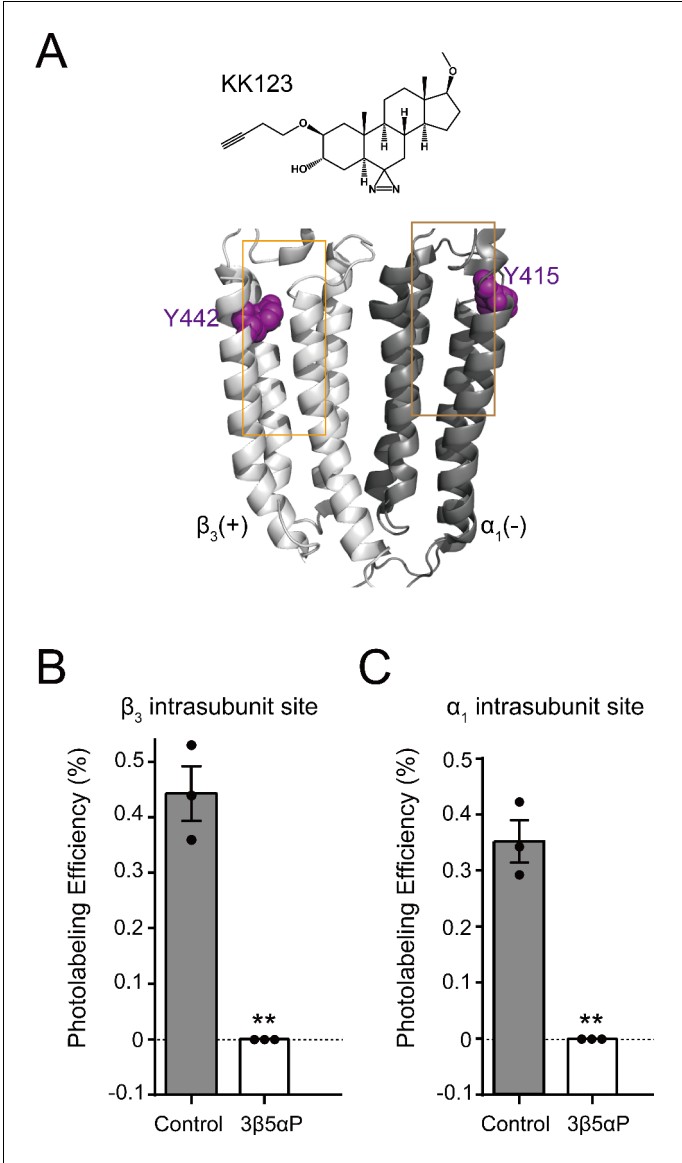

**Figure 5.** Epi-allopregnanolone prevents neurosteroid photolabeling at the α$_1$ and β$_3$ intrasubunit sites. (**A**) Structures of the neurosteroid photolabeling reagent KK123 and the α$_1$β$_3$ GABA$_A$R-TMDs highlighting the residues Y442 in the β$_3$ intrasubunit site and Y415 in the α$_1$ intrasubunit site previously identified by KK123 photolabeling in purple. Adjacent β$_3$(+) and α$_1$(-) subunits are shown and the channel pore is behind the subunits. (**B**) Photolabeling efficiency of β$_3$ subunit TM4 (β$_3$ intrasubunit site) in α$_1$β$_3$ GABA$_A$R by 3 μM KK123 in the absence or presence of 30 μM epi-allopregnanolone (3β5αP). Statistical differences are compared using unpaired *t*-test (*n* = 3,± SEM). **p<0.01 vs. control. (**C**) Same as (**B**) for α$_1$ subunit TM4 (α$_1$ intrasubunit site, *n* = 3,± SEM).

the conditions used in the [³H]muscimol binding assays and photolabeled residues were identified and labeling efficiency was determined using middle-down mass spectrometry (*Chen et al., 2019*). KK148, KK150, 3α5αP and 3β5αP all prevented KK200 photolabeling of $\alpha_1$N408 in the $\alpha_1$ intrasubunit site (*Figure 4B*), consistent with their binding to this site. In contrast, KK148, KK150 and 3α5αP but not 3β5αP prevented labeling of $\beta_3$G308 in the intersubunit site (*Figure 4C*), indicating that 3β5αP does not bind to the intersubunit site. Similarly, 3β5αP did not prevent labeling of the intersubunit site by a similar NS-analogue photolabeling reagent in detergent-solubilized GABA$_A$Rs (*Jayakar et al., 2020*).

The KK148- and KK150-photolabeled (30 µM) samples were also analyzed to directly identify the sites of adduction. In both the KK148- and KK150-labeled samples, photolabeled peptides were identified from the TM4 helices of both the $\alpha_1$ and $\beta_3$ subunits. The labeled peptides had longer chromatographic elution times than the corresponding unlabeled peptides and corresponded with high mass accuracy (<20 ppm) to the predicted mass of the unlabeled peptides plus the add weight minus N$_2$ of KK148 or KK150 (*Figure 4—figure supplement 1*). Product ion (MS2) spectra of the KK148- and KK150-labeled peptides from the $\alpha_1$ subunit identified the labeled residue as Y415 for both KK148 and KK150 with photolabeling efficiencies of 0.77% and 0.62%, respectively (*Figure 4D–E*, *Figure 4—figure supplement 2*); Y415 is the same residue labeled by KK123 at the $\alpha_1$ intrasubunit site (*Chen et al., 2019*). The KK148 and KK150 labeled peptides in TM4 of the $\beta_3$ subunit and corresponding unlabeled peptide were identified by fragmentation spectra as $\beta_3$TM4 I426-N445. These data support labeling of the $\beta_3$ intrasubunit site by KK148 and KK150. Fragmentation spectra of the peptide-sterol adducts were not adequate to determine the precise labeled residue because of low photolabeling efficiency (0.13% for KK148; 0.19% for KK150, *Figure 4—figure supplement 1*). No photolabeled peptides were identified in the $\beta_3$(+)–$\alpha_1$(-) intersubunit site. This is likely because KK148 and KK150, similar to KK123, utilize an aliphatic diazirine that preferentially labels nucleophilic residues (*Sugasawa et al., 2019*; *Budelier et al., 2017*; *Das, 2011*); such residues are not present in the intersubunit site.

We have also shown that KK123 labeling of the $\alpha_1$ intrasubunit ($\alpha_1$Y415) and $\beta_3$ intrasubunit ($\beta_3$Y442) sites (*Figure 5A*) can be prevented by a 10fold excess of 3α5αP (*Chen et al., 2019*). We thus examined whether 3β5αP (30 µM) inhibited photolabeling by KK123 (3 µM). 3β5αP completely inhibited KK123 photolabeling at both intrasubunit sites (*Figure 5B–C*). Collectively, the data show that KK148, KK150 and 3α5αP bind to all three of the identified NS-binding sites. In contrast, 3β5αP selectively binds to the two intrasubunit binding sites, but not to the canonical $\beta_3$(+)–$\alpha_1$(-) intersubunit site.

## Orthosteric ligand binding enhancement by NS analogues is mediated by distinct sites

To determine which of the previously identified binding sites contributes to NS enhancement of [³H] muscimol binding, we performed site-directed mutagenesis of the NS-binding sites previously determined by photolabeling (*Figure 6A*; *Chen et al., 2019*). Specifically, $\alpha_1$(Q242L)$\beta_3$ targets the $\beta_3$(+)–$\alpha_1$(-) intersubunit site, $\alpha_1$(N408A/Y411F)$\beta_3$ and $\alpha_1$(V227W)$\beta_3$ the $\alpha_1$ intrasubunit site, and $\alpha_1\beta_3$(Y284F) the $\beta_3$ intrasubunit site. None of these mutations produced a significant change in [³H] muscimol K$_d$ (*Figure 7B* and *Figure 7—source data 1*). Accordingly, concentration-dependent NS effects were assayed at a fixed concentration of [³H]muscimol (3 nM; ~EC$_5$). It should be noted that earlier studies showed two-component binding curves for [³H]muscimol in brain membranes, with NS causing an increase in the B$_{max}$ of the high-affinity component (*Harrison and Simmonds, 1984*). In contrast, our results with expressed $\alpha_1\beta_3$ GABA$_A$Rs show a single-component [³H]muscimol binding curve with NS producing an increase in muscimol affinity. Our results are similar to results reported with expressed $\alpha_1\beta_3\gamma_2$ GABA$_A$Rs, where allosteric modulators increased the affinity of a single-component [³H]muscimol binding curve (*Dostalova et al., 2014*). Whether the complex [³H]muscimol binding curves observed in brain is the result of heterogeneity of receptor subtypes or multiple states of the GABA$_A$R is unresolved.

Mutations in the $\beta_3$(+)–$\alpha_1$(-) intersubunit and $\alpha_1$ intrasubunit sites decreased 3α5αP enhancement of [³H]muscimol binding by ~80%, while mutation of the $\beta_3$ intrasubunit site led to a small decrease (*Figure 6B*, *Table 1*). The residual enhancement of [³H]muscimol binding observed in receptors with mutations in the intersubunit or $\alpha_1$ intrasubunit site occurs at 10fold higher concentrations of 3α5αP than wild-type (WT) and receptors with mutations in the $\beta_3$ intrasubunit site (*Table 1*), suggesting

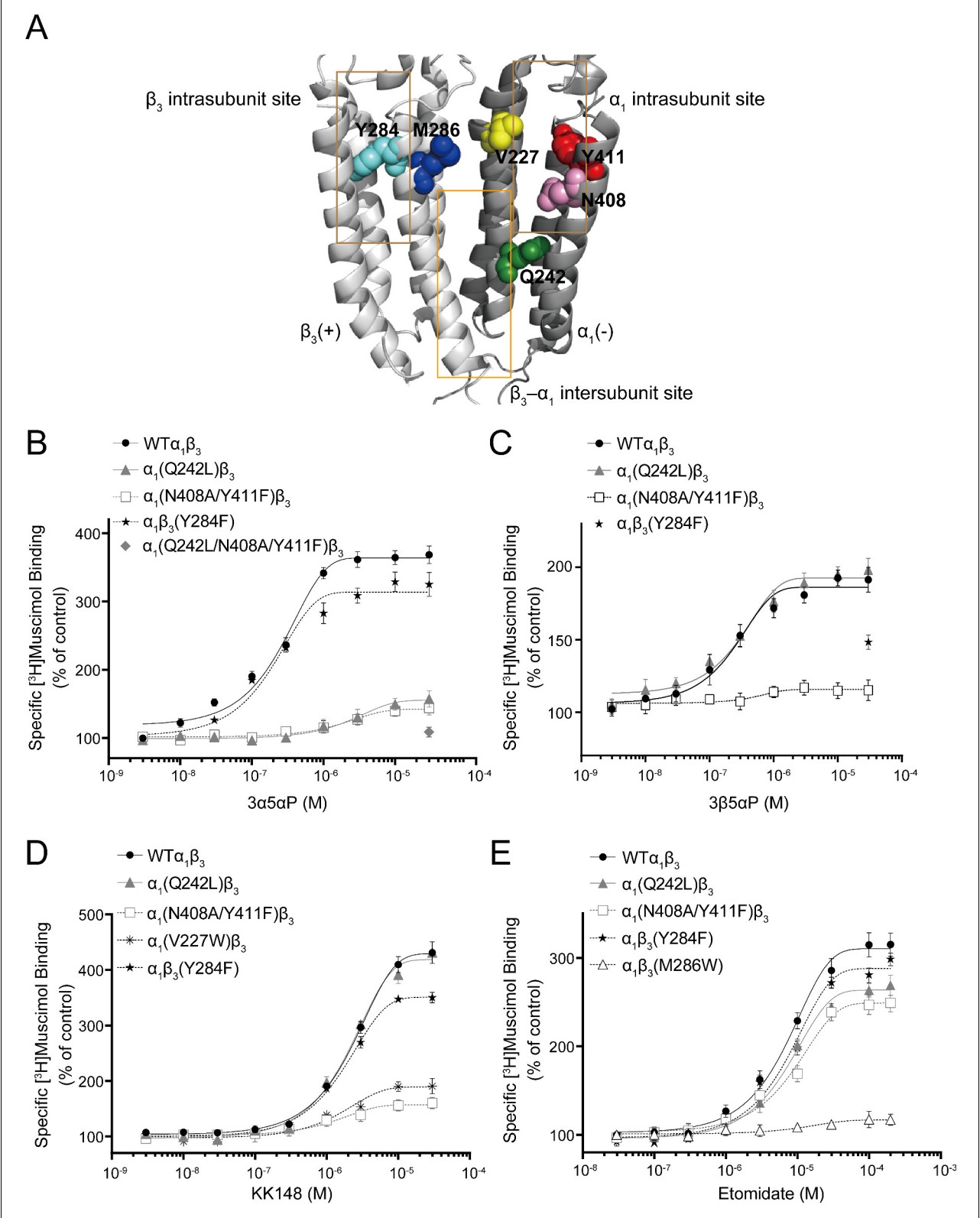

**Figure 6.** Effect of mutations in $\alpha_1\beta_3$ GABA$_A$R on neurosteroid modulation of [$^3$H]muscimol binding. (**A**) Structure of the $\alpha_1\beta_3$ GABA$_A$R-TMD highlighting the residues where mutations were made in putative binding sites for neurosteroids (Q242-green for $\beta_3$–$\alpha_1$ intersubunit site; V227-yellow, N408-pink and Y411-red for $\alpha_1$ intrasubunit site; Y284-cyan for $\beta_3$ intrasubunit site) and M286-blue for etomidate. Adjacent $\beta_3$(+) and $\alpha_1$(-) subunits are shown and the channel pore is behind the subunits. (**B**) Concentration-response relationship for the effect of 3 nM–30 µM allopregnanolone (3$\alpha$5$\alpha$P) on

*Figure 6 continued on next page*

Figure 6 continued

[³H]muscimol (3 nM) binding to $\alpha_1\beta_3$ GABA$_A$R WT and indicated mutants. Data points represent mean ± SEM ($n$ = 6). (C), (D) and (E) Same as (B) for 3 nM–30 μM epi-allopregnanolone (3β5αP) ($n$ = 3), KK148 ($n$ = 6) and 30 nM–200 μM etomidate ($n$ = 6), respectively. The data for WT in panels 6B and 6D is a replot of the same data shown in *Figure 1E*.

The online version of this article includes the following figure supplement(s) for figure 6:

**Figure supplement 1.** Time course of neurosteroid modulation of muscimol binding.

that 3α5αP binds to the $\beta_3$ intrasubunit site with lower affinity. In contrast, mutations in the $\alpha_1$ and $\beta_3$ intrasubunit sites, but not the intersubunit site decreased the enhancement of [³H]muscimol binding by 3β5αP and KK148 (*Figure 6C–D*, *Table 1*). To confirm that the effect of these mutations on NS effect are steroid-specific, we also tested their effect on etomidate, which enhances [³H]muscimol binding in $\alpha_1\beta_3$ GABA$_A$Rs and acts through a binding site distinct from NS (*Li et al., 2006*; *Jayakar et al., 2019*). The mutations targeting NS-binding sites resulted in modest decreases in [³H]muscimol binding enhancement by etomidate; however, the $\alpha_1\beta_3$(M286W) mutation which abolishes etomidate potentiation and activation of GABA$_A$Rs (*Stewart et al., 2008*; *Ziemba et al., 2018*), also abolished [³H]muscimol binding enhancement (*Figure 6E*).

We did not test the effects of mutations on KK150 action because it minimally enhances [³H]muscimol binding. However, KK150 binds to all three of the identified NS-binding sites, and may thus be a weak partial agonist or antagonist at the sites mediating NS enhancement of [³H]muscimol binding. Consistent with this prediction, KK150 inhibited enhancement of [³H]muscimol binding by 3α5αP and KK148 (*Figure 8*).

Collectively, these results show that multiple NS-binding sites contribute to enhancement of [³H]muscimol affinity and that potentiating NS (3α5αP) and non-potentiating NS (3β5αP, KK148 and KK150) have both common and distinct sites of action. Specifically, 3α5αP enhances [³H]muscimol binding through all three sites but predominantly through the intersubunit and $\alpha_1$ intrasubunit sites, which we have previously shown mediate PAM-NS potentiation (*Chen et al., 2019*). In contrast, 3β5αP and KK148 enhance [³H]muscimol binding exclusively through the $\alpha_1$ and $\beta_3$ intrasubunit sites. KK150 antagonizes the effects of KK148 on [³H]muscimol binding, presumably via the intrasubunit sites, and antagonizes the effects of 3α5αP, possibly via all three sites. These data indicate that NS binding to both the intersubunit and intrasubunit sites contributes to 3α5αP enhancement of [³H] muscimol binding, but that only the intrasubunit binding sites contribute to the effects of 3β5αP and KK148. The data (*Figure 6B–D*) are consistent with NS producing their effects by independent action at each of the binding sites and our interpretation is based on that assumption. We cannot, however, rule out the possibility of an allosteric interaction between the NS binding sites (*Chen et al., 2019*).

It is important to note that the [³H]muscimol binding curves in *Figure 6* are normalized to control. The raw data show that membranes containing WT receptors have 10–20-fold higher [³H]muscimol binding (B$_{max}$) than membranes containing $\alpha_1$(N408A/Y411F)$\beta_3$ receptors (*Figure 7A*), whereas the B$_{max}$ of membranes containing $\alpha_1$(Q242L)$\beta_3$, $\alpha_1\beta_3$(Y284F) or $\alpha_1$(V227W)$\beta_3$ receptors was the same as for WT $\alpha_1\beta_3$ GABA$_A$Rs (*Figure 7—source data 1*). The lower total [³H]muscimol binding observed in $\alpha_1$(N408A/Y411F)$\beta_3$ membranes is likely a consequence of decreased receptor expression. To assure that differences in NS effect between WT and $\alpha_1$(N408A/Y411F)$\beta_3$ are not due to different muscimol affinities, we examined [³H]muscimol binding at a full range of concentrations. The $\alpha_1$(N408A/Y411F)$\beta_3$ mutations did not have a significant effect on [³H]muscimol affinity (*Figure 7B*), but eliminated the modulatory effects of NS (3α5αP and KK148) on [³H]muscimol affinity (*Figure 7C–D* and *Figure 7—source data 2*). To assure that the effect of $\alpha_1$(N408A/Y411F)$\beta_3$ was specific to NS, we also examined the effect of etomidate (a non-steroidal GABA$_A$R PAM) on muscimol affinity. Etomidate enhanced [³H]muscimol affinity in both the WT and $\alpha_1$(N408A/Y411F)$\beta_3$ receptors, indicating that the effect of these mutations are specific to NS action (*Figure 7C–D*).

## 3β5αP increases desensitization through binding to $\alpha_1$ and $\beta_3$ intrasubunit sites

To further explore the relationship between desensitization and enhancement of [³H]muscimol binding, we examined the consequences of mutations to these sites on physiological measurements of desensitization induced by NS. Again, these experiments were performed with 3β5αP because it is

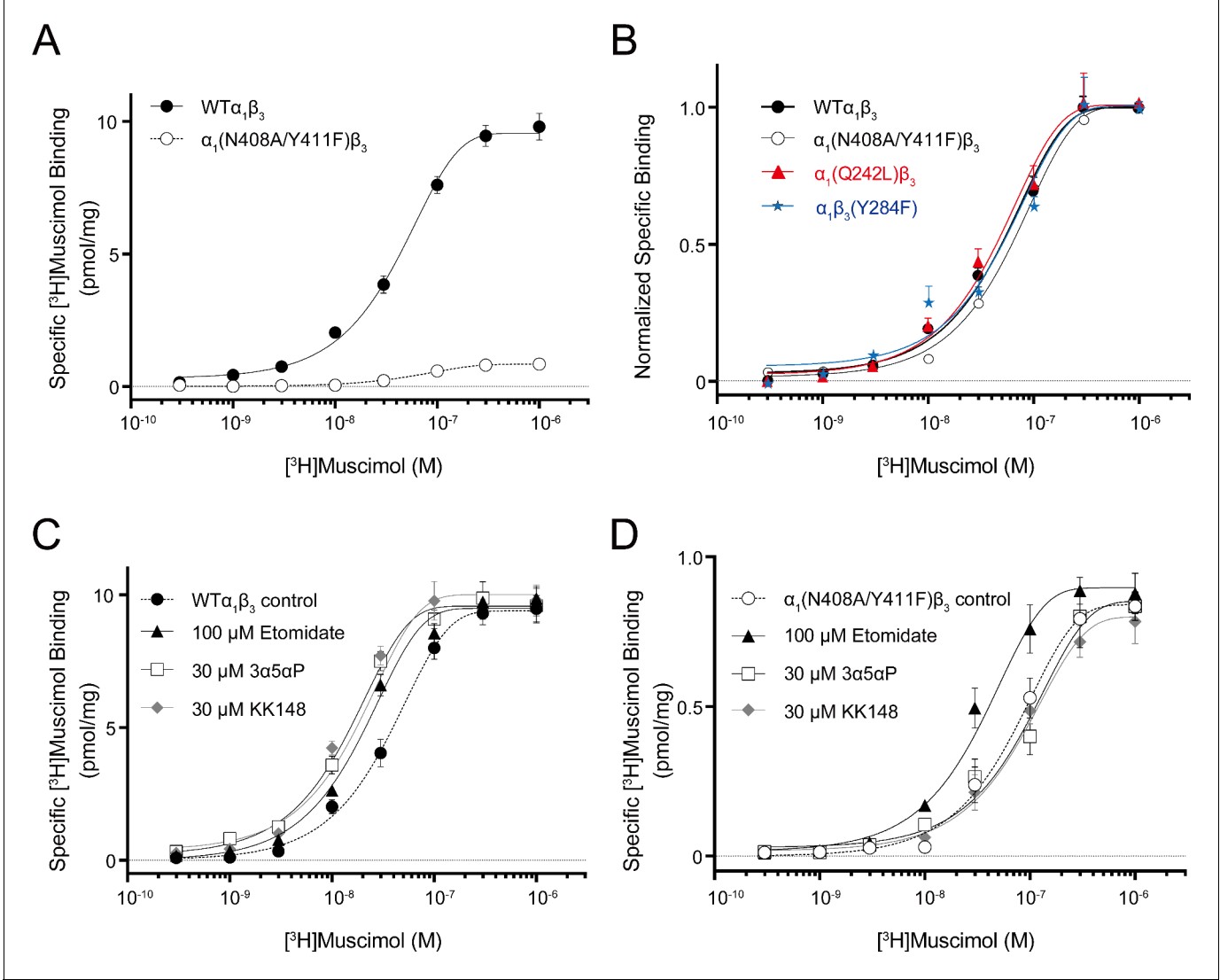

**Figure 7.** Neurosteroid effect on [³H]muscimol binding isotherms in $\alpha_1\beta_3$ WT and $\alpha_1$(N408A/Y411F)$\beta_3$ GABA$_A$Rs. (**A**) [³H]muscimol binding isotherms (0.3 nM–1 μM) for $\alpha_1\beta_3$ GABA$_A$R WT and $\alpha_1$(N408A/Y411F)$\beta_3$ GABA$_A$R. Data points are presented as mean ± SEM ($n$ = 3). (**B**) Normalized curves of [³H] muscimol binding isotherms (0.3 nM–1 μM) for $\alpha_1\beta_3$ GABA$_A$R WT and representative mutated receptors for each neurosteroid binding site [i.e. $\alpha_1$(Q242L)$\beta_3$ for the $\beta_3$–$\alpha_1$ intersubunit site; $\alpha_1$(N408A/Y411F)$\beta_3$ for the $\alpha_1$ intrasubunit site; $\alpha_1\beta_3$(Y284F) for the $\beta_3$ intrasubunit site]. Each data point represents mean ± SEM ($n$ = 6 for WT; $n$ = 3 for mutated receptors). (**C**) Effect of 100 μM etomidate, 30 μM allopregnanolone (3$\alpha$5$\alpha$P) and 30 μM KK148 on [³H]muscimol binding isotherms in the $\alpha_1\beta_3$ GABA$_A$R WT. (**D**) Same as (**C**) in the $\alpha_1$(N408A/Y411F)$\beta_3$ mutant. Each data point represents mean ± SEM ($n$ = 3).

The online version of this article includes the following source data and figure supplement(s) for figure 7:

**Source data 1.** Properties of [³H]muscimol binding isotherms in $\alpha_1\beta_3$ WT and mutant GABA$_A$Rs.

**Source data 2.** Properties of neurosteroid effect on [³H]muscimol binding isotherms in $\alpha_1\beta_3$ GABA$_A$R WT and $\alpha_1$(N408A/Y411F)$\beta_3$ mutant.

**Figure supplement 1.** Total, nonspecific and specific [³H]muscimol binding curves.

the endogenous 3β-OH NS and because of limited quantities of KK148. Desensitization was quantified by defining the baseline steady-state current at 1 mM GABA as 100% and measuring percent reduction of the steady-state current elicited by a NS (*Figure 9A* inset). While GABA was at a saturating concentration for all receptors, the peak $P_{open}$ it elicited was <<1.0 for several of the mutated receptors. To normalize $P_{open}$, mutated receptors with low peak $P_{open}$ were activated by co-application of 1 mM GABA with 40 μM pentobarbital (PB) (*Steinbach and Akk, 2001*) prior to application of 3$\alpha$5$\alpha$P. This was done because the magnitude of negative allosteric modulation varies as a function of $P_{open}$ (*Germann et al., 2019a*). To ensure that PB did not influence NS negative allosteric

**Table 1.** Effects of mutations on neurosteroid modulation of [$^3$H]muscimol binding.

$EC_{50}$, Hill slope and maximal effect values [$E_{max}$ (% of control): 100% means no effect] for the concentration-response curves in **Figure 6B–E**. Statistical differences are analyzed using one-way ANOVA with Bonferroni's multiple comparisons test (*$p < 0.05$ vs. WT; **$p < 0.01$ vs. WT). Data are presented as mean ± SEM.

| 3α5αP | $EC_{50}$ (μM) | Hill slope | $E_{max}$ (% of control) | N |
|---|---|---|---|---|
| WTα$_1$β$_3$ | 0.24 ± 0.04 | 1.10 ± 0.14 | 374.1 ± 11.1 | 6 |
| α$_1$(Q242L)β$_3$ | **2.66 ± 0.51 | 1.16 ± 0.37 | **159.8 ± 10.9 | 6 |
| α$_1$(N408A/Y411F)β$_3$ | **2.30 ± 0.48 | 0.87 ± 0.44 | **146.0 ± 9.3 | 6 |
| α$_1$β$_3$(Y284F) | 0.19 ± 0.04 | 0.87 ± 0.16 | 342.3 ± 13.9 | 6 |
| α$_1$(Q242L/N408A/Y411F)β$_3$ | - | - | **105.9 ± 7.3 | 6 |
| **3β5αP** | | | | |
| WTα$_1$β$_3$ | 0.25 ± 0.08 | 0.84 ± 0.23 | 195.1 ± 6.7 | 3 |
| α$_1$(Q242L)β$_3$ | 0.27 ± 0.09 | 0.77 ± 0.21 | 204.3 ± 4.5 | 3 |
| α$_1$(N408A/Y411F)β$_3$ | 0.61 ± 0.26 | 2.25 ± 0.92 | **124.3 ± 2.6 | 3 |
| α$_1$β$_3$(Y284F) | - | - | **148.6 ± 4.9 | 3 |
| **KK148** | | | | |
| WTα$_1$β$_3$ | 2.40 ± 0.36 | 1.36 ± 0.16 | 431.0 ± 19.5 | 6 |
| α$_1$(Q242L)β$_3$ | 2.20 ± 0.31 | 1.24 ± 0.12 | 434.5 ± 5.6 | 6 |
| α$_1$(N408A/Y411F)β$_3$ | 1.63 ± 0.53 | 0.73 ± 0.23 | **161.7 ± 3.5 | 6 |
| α$_1$(V227W)β$_3$ | 1.73 ± 0.68 | 0.76 ± 0.31 | **209.2 ± 7.4 | 6 |
| α$_1$β$_3$(Y284F) | 1.79 ± 0.44 | 1.35 ± 0.13 | **357.2 ± 8.1 | 6 |
| **Etomidate** | | | | |
| WTα$_1$β$_3$ | 7.24 ± 1.18 | 1.07 ± 0.17 | 331.1 ± 9.9 | 6 |
| α$_1$(Q242L)β$_3$ | 7.50 ± 0.95 | 1.35 ± 0.20 | **277.8 ± 10.9 | 6 |
| α$_1$(N408A/Y411F)β$_3$ | 9.14 ± 2.20 | 1.07 ± 0.26 | **268.2 ± 5.9 | 6 |
| α$_1$β$_3$(Y284F) | 7.71 ± 1.10 | 0.90 ± 0.11 | 303.5 ± 5.8 | 6 |
| α$_1$β$_3$(M286W) | *22.5 ± 6.17 | 0.50 ± 0.16 | **128.6 ± 7.8 | 6 |

modulation, control experiments were performed in WT α$_1$β$_3$ GABA$_A$Rs and showed no significant difference in the desensitization elicited by 3β5αP between receptors activated by GABA vs. GABA plus PB. The maximum P$_{open}$ for the mutant receptors varied between 0.55 and 0.95; some of the NS effects on macroscopic currents may be influenced by these differences.

3β5αP reduced the steady-state current (i.e. enhanced desensitization) by 23.0 ± 2.8% (% of desensitization: mean ± SEM, $n = 5$, **Figure 9A**). Mutations in the α$_1$ and β$_3$ intrasubunit sites [i.e. α$_1$(N408A/Y411F)β$_3$ and α$_1$β$_3$(Y284F), respectively] prevented 3β5αP-enhanced desensitization by ~67% (**Figure 9B**), whereas mutation in the β(+)–α(-) intersubunit site [α$_1$(Q242L)β$_3$] was without effect (**Figure 9B**). Receptors with mutations in both the α$_1$ and β$_3$ intrasubunit sites [α$_1$(N408A/Y411F)β$_3$(Y284F)] showed less NS-enhancement of desensitization than receptors with mutations in either of the intrasubunit sites alone, indicating that both intrasubunit sites contribute to the desensitizing effect (**Figure 9B**). Although the desensitizing effect of 3β5αP is completely eliminated by the V2′S mutation α$_1$(V256S)β$_3$, it is not completely eliminated by combined mutations of all three binding sites [α$_1$(Q242L/N408A/Y411F)β$_3$(Y284F)] (**Figure 9B**). This suggests either that the effects of the mutations are incomplete or there are additional unidentified NS-binding sites contributing to desensitization. Since mutations of the α$_1$ and β$_3$ intrasubunit sites also disrupt 3β5αP-enhancement of [$^3$H]muscimol binding (**Figure 6C**), we conclude that 3β5αP binding to these intrasubunit sites stabilizes the desensitized state of the GABA$_A$R and thus enhances [$^3$H]muscimol binding. Furthermore, KK148 increased GABA$_A$R desensitization (% of desensitization = 27.2 ± 6.0: mean ± SEM, $n = 3$, **Figure 2A**) and the α$_1$(V256S)β$_3$ mutation abolished the effect (% of desensitization = 0, $n = 1$). These observations support the idea that binding of certain NS analogues to α$_1$ and β$_3$ intrasubunit

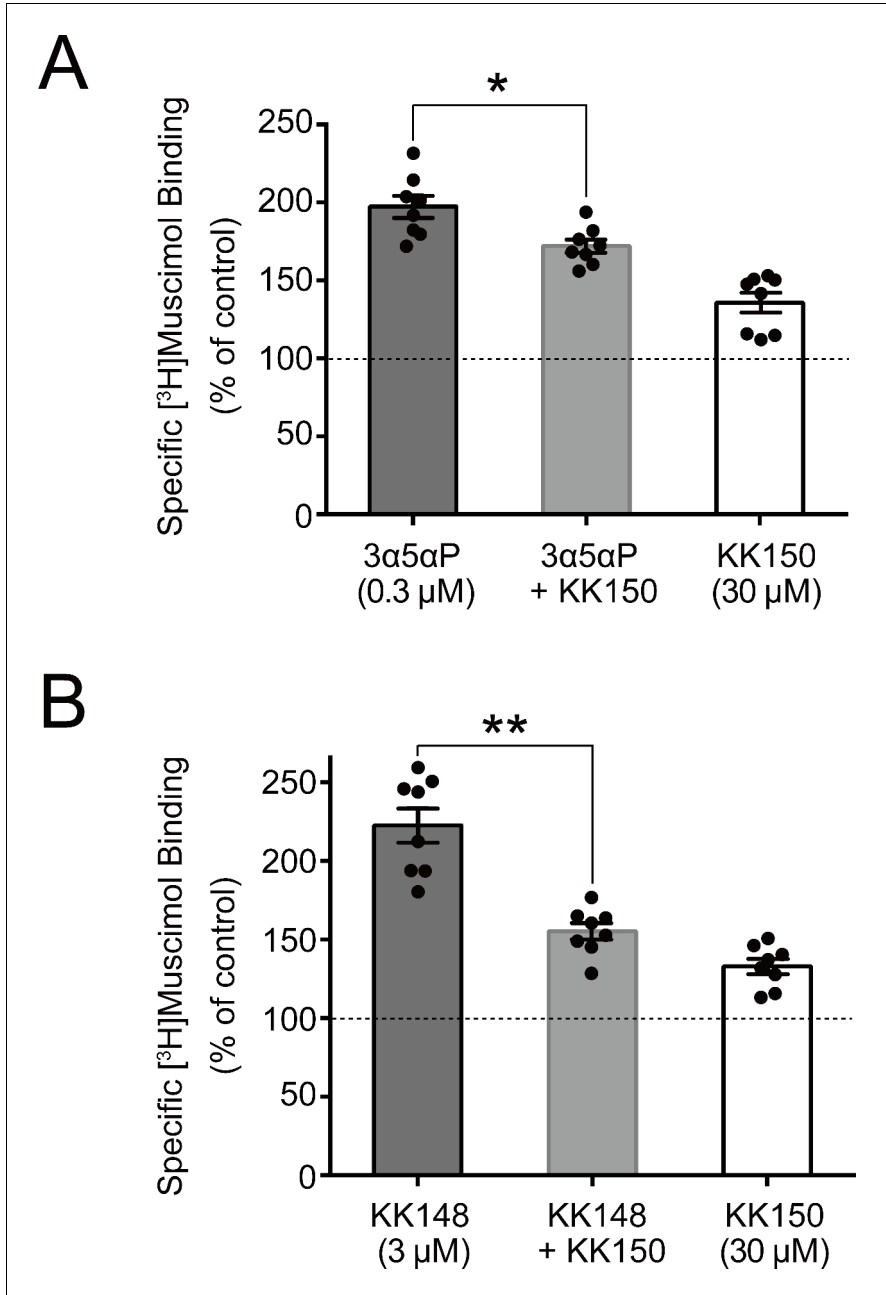

**Figure 8.** KK150 prevents neurosteroid-induced muscimol binding enhancement. (**A**) Enhancement of specific [$^3$H] muscimol (3 nM) binding to $\alpha_1\beta_3$ GABA$_A$R by 0.3 μM allopregnanolone (3α5αP) in the absence (black bar) or presence (grey bar) of 30 μM KK150 and KK150 alone (white bar). Statistical differences are analyzed using one-way ANOVA with Bonferroni's multiple comparisons test ($n = 8$, ± SEM). *p<0.05 vs. 0.3 μM 3α5αP alone. (**B**) Same as (**A**) for 3 μM KK148 ($n = 8$, ± SEM). **p<0.01 vs. 3 μM KK148 alone.

sites, increases GABA$_A$R desensitization. In contrast, KK150 showed a very small effect on desensitization (% of desensitization = 2.1 ± 0.7: mean ± SEM, $n = 5$, *Figure 2B*), consistent with the small increase in [$^3$H]muscimol binding by KK150 (*Figure 1E*).

## The effects of 3α5αP binding to intrasubunit sites on desensitization

3α5αP binds to all three of the NS-binding sites on $\alpha_1\beta_3$ GABA$_A$R, and mutations in all three sites reduce 3α5αP enhancement of [$^3$H]muscimol binding (*Figure 6B*). This suggests the possibility that activation by 3α5αP (mediated primarily by the $\beta_3$(+)–$\alpha_1$(-) intersubunit site) masks a desensitizing

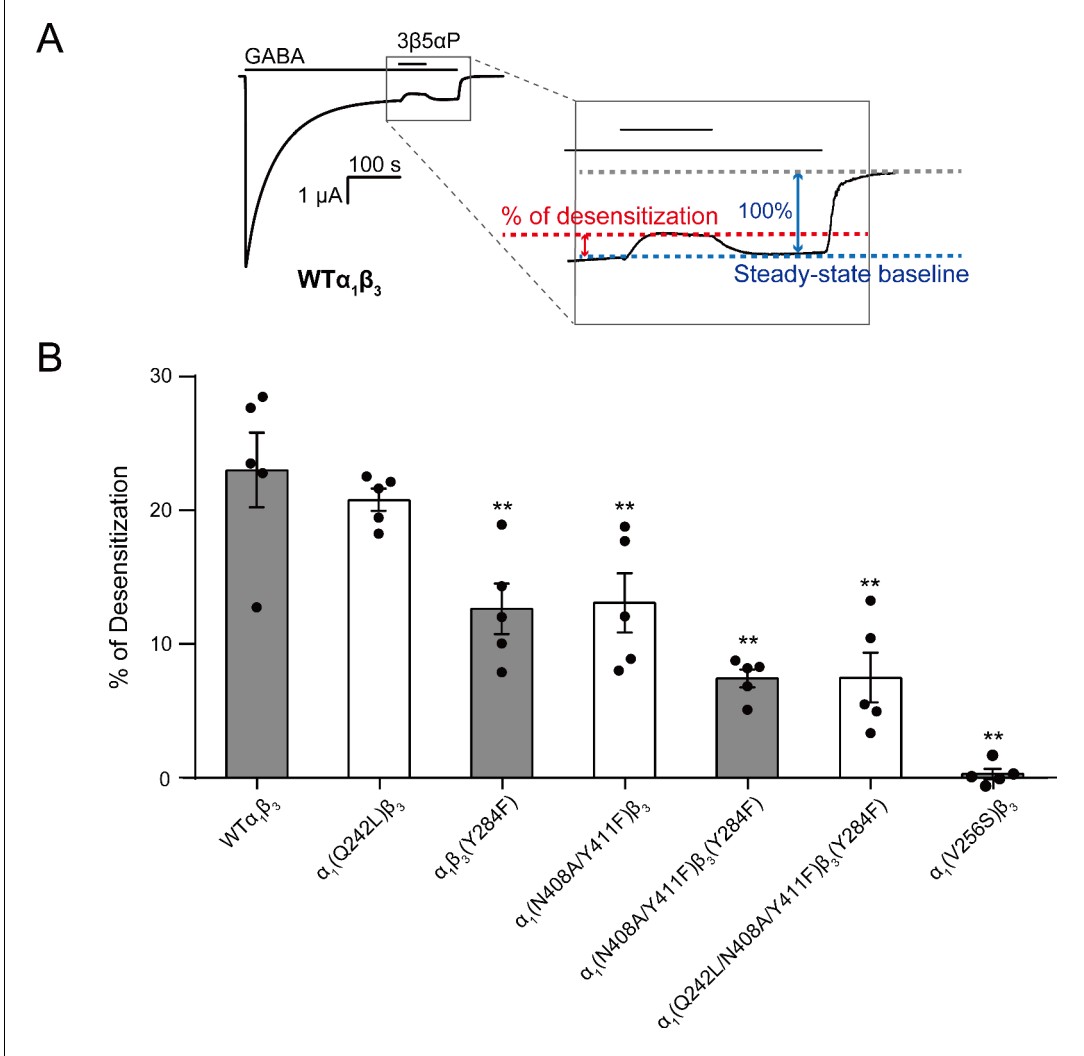

**Figure 9.** Mutations in intrasubunit sites prevent desensitization by epi-allopregnanolone. (**A**) Sample current trace showing the effect of 3 µM epi-allopregnanolone (3β5αP) on steady-state current elicited by continuous administration of 1 mM GABA to $\alpha_1\beta_3$ GABA$_A$R expressed in oocytes. A zoomed-in box shows neurosteroid-induced desensitization of the steady-state GABA current. (**B**) Percent desensitization of the steady-state $\alpha_1\beta_3$ GABA$_A$R currents (WT and mutants) by 3 µM 3β5αP during continuous application of 1 mM GABA [for WT, $\alpha_1$(Q242L)$\beta_3$, $\alpha_1\beta_3$(Y284F), $\alpha_1$(N408A/Y411F)$\beta_3$ and $\alpha_1$(V256S)$\beta_3$ GABA$_A$Rs] or 1 mM GABA + 25 µM pentobarbital (PB) [for $\alpha_1$(N408A/Y411F)$\beta_3$(Y284F) and $\alpha_1$(Q242L/N408A/Y411F)$\beta_3$(Y284F) GABA$_A$Rs]. The combination of GABA and PB is essential for some mutated receptors to obtain a high, consistent peak open probability. Statistical differences are analyzed using one-way ANOVA with Bonferroni's multiple comparisons test ($n$ = 5,± SEM). \*\*p<0.01 vs. WT.

effect mediated through the $\beta_3$ and/or $\alpha_1$ intrasubunit binding sites. To determine whether intrasubunit binding sites mediate increased desensitization by 3α5αP, we examined the effect of 3α5αP on steady-state currents in receptors with mutations in the $\alpha_1$ or $\beta_3$ intrasubunit site. Mutations in the intrasubunit sites were prepared with a background $\alpha_1$(Q242L)$\beta_3$ mutation to remove 3α5αP activation (*Chen et al., 2019*; *Sugasawa et al., 2019*; *Akk et al., 2008*; *Bracamontes and Steinbach, 2009*) and focus on the effects of 3α5αP on the equilibrium between the open and desensitized states.

3α5αP produced a small reduction in steady-state current in $\alpha_1$(Q242L)$\beta_3$ receptors with mutations in neither of the intrasubunit sites (*Figure 10A*). This inhibitory effect was eliminated by $\alpha_1$(V256S)$\beta_3$, indicating that it was due to receptor desensitization (*Figure 10D*). In receptors with combined mutations in the intersubunit and $\alpha_1$ intrasubunit sites [i.e. $\alpha_1$(Q242L/N408A/Y411F)$\beta_3$], 3α5αP significantly inhibited the steady-state current (*Figure 10B*), an effect that was markedly reduced by mutations in the $\beta_3$ intrasubunit site [$\alpha_1$(Q242L)$\beta_3$(Y284F)] (*Figure 10C*). These data

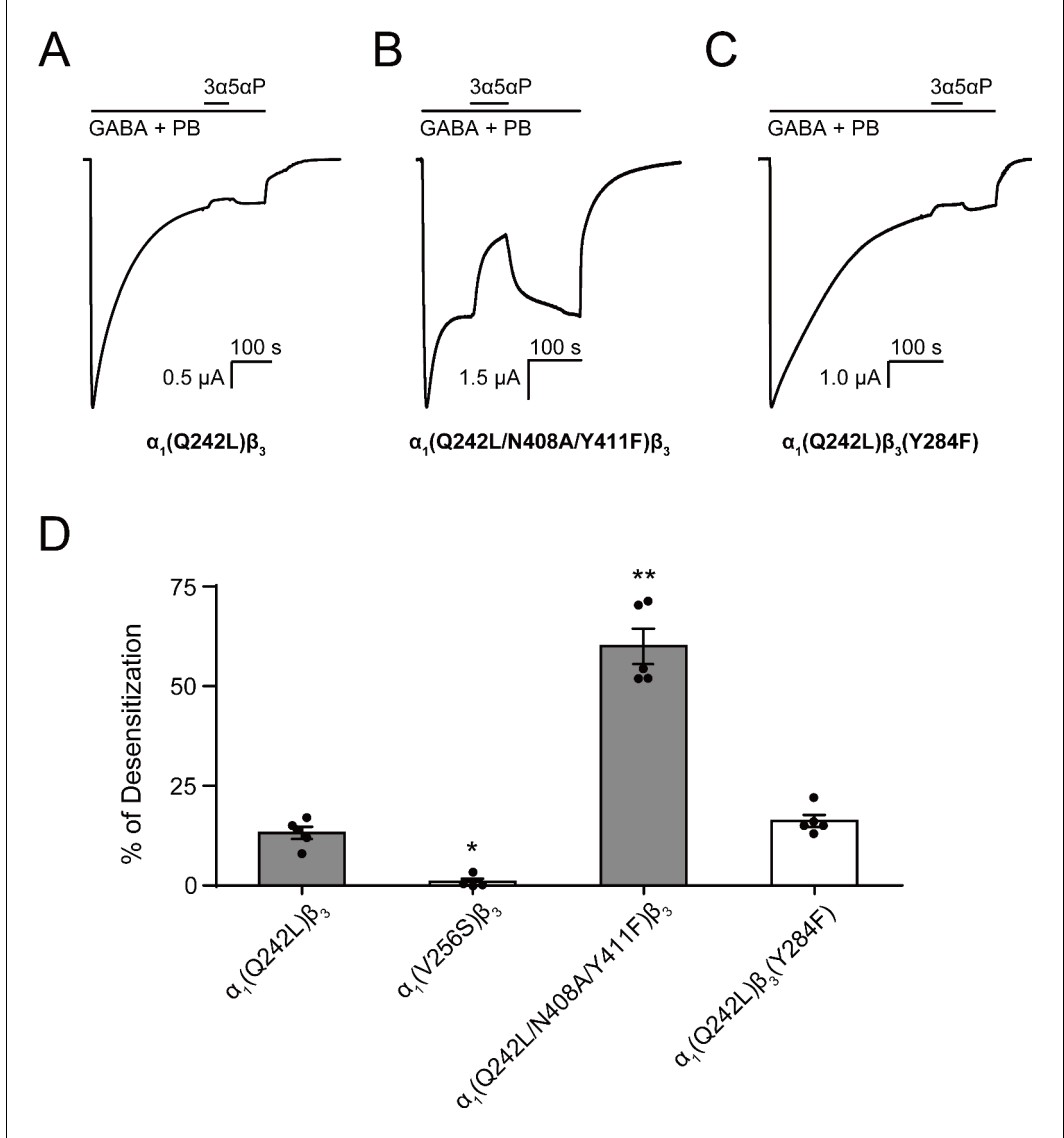

**Figure 10.** Allopregnanolone desensitizes GABA$_A$R currents via binding to the β$_3$ intrasubunit site. (**A**) Sample current trace showing the effect of 3 μM allopregnanolone (3α5αP) on α$_1$(Q242L)β$_3$ GABA$_A$R activated by 1 mM GABA co-applied with 40 μM pentobarbital (PB). (**B**), (**C**) Same as (**A**) for α$_1$(Q242L/N408A/Y411F)β$_3$ GABA$_A$R and α$_1$(Q242L)β$_3$(Y284F) GABA$_A$R, respectively. (**D**) Percent desensitization of the steady-state currents elicited by 1 mM GABA with 40 μM PB in α$_1$β$_3$ GABA$_A$R with specified mutations. Statistical differences are analyzed using one-way ANOVA with Bonferroni's multiple comparisons test [$n$ = 4 for α$_1$(V256S)β$_3$; $n$ = 5 for others,± SEM]. *$p < 0.05$; **$p < 0.01$ vs. α$_1$(Q242L)β$_3$, respectively.

suggest that 3α5αP exerts a desensitizing effect by binding to the β$_3$ intrasubunit site and that 3α5αP binding to the α$_1$ intrasubunit site does not promote desensitization (*Figure 10D*). Notably, 3α5αP exerted only a modest inhibitory effect in α$_1$(Q242L)β$_3$ receptors in which occupancy of the β$_3$ intrasubunit site should promote inhibition. This may be due to a counterbalancing action at the α$_1$ intrasubunit site, where 3α5αP binding contributes more to receptor activation as demonstrated by our previous observation that mutations in the α$_1$ intrasubunit site significantly reduce 3α5αP potentiation of GABA-elicited currents (*Chen et al., 2019*). These results suggest that in addition to activation, 3α5αP enhances receptor desensitization. Enhanced desensitization by the PAM-NS 3α5αP (*Haage and Johansson, 1999*) and 3α5α-THDOC (*Zhu and Vicini, 1997*; *Bianchi and Macdonald, 2003*) has been observed in prior studies supporting the current finding with 3α5αP.

# Discussion

In this study, we examined how site-specific binding to the three identified NS sites on $\alpha_1\beta_3$ GABA$_A$R (*Chen et al., 2019*) contributes to the PAM vs. NAM activity of epimeric 3-OH NS. We found that the PAM-NS $3\alpha5\alpha$P, but not the NAM-NS $3\beta5\alpha$P, binds to the canonical $\beta_3$(+)–$\alpha_1$(-) intersubunit site that mediates receptor potentiation, explaining the absence of $3\beta5\alpha$P PAM activity. In contrast, $3\beta5\alpha$P binds to intrasubunit sites in the $\alpha_1$ and $\beta_3$ subunits, promoting receptor desensitization. Binding to the intrasubunit sites provides a mechanistic explanation for the NAM effects of $3\beta5\alpha$P (*Wang et al., 2002*). $3\alpha5\alpha$P also binds to the $\beta_3$ intrasubunit site explaining the previously described desensitizing effect of the PAM-NS $3\alpha5\alpha$P (*Haage and Johansson, 1999*) and $3\alpha5\alpha$-THDOC (*Zhu and Vicini, 1997*; *Bianchi and Macdonald, 2003*). Two synthetic NS with diazirine moieties at C3 (KK148 and KK150) were used to identify NS-binding sites and shown to bind to the intersubunit as well as both intrasubunit sites. Neither of these ligands potentiated agonist-activated GABA$_A$R currents, reinforcing the importance of the 3$\alpha$-OH group and its interaction with $\alpha_1$Q242 in PAM actions. KK148 is an efficacious desensitizing agent, acting through the $\alpha_1$ and $\beta_3$ intrasubunit NS-binding sites. KK150, the 17$\alpha$-epimer of KK148, binds to all three NS-binding sites, but neither activates nor desensitizes GABA$_A$Rs, suggesting a potential chemical scaffold for a general NS antagonist. Collectively, these data show that differential occupancy of and efficacy at three discrete NS-binding sites determines whether a NS ligand has PAM, NAM, or potentially NS antagonist activity on GABA$_A$Rs.

The observation that $3\beta5\alpha$P and KK148 enhance orthosteric ligand binding but do not potentiate GABA-elicited currents first suggested that these NAM-NS selectively stabilize a non-conducting state that has high affinity to the orthosteric agonist muscimol. This liganded/closed state could represent a pre-active (*Gielen and Corringer, 2018*) or a desensitized conformation of the receptor (n.b. there may be multiple desensitized conformations of the receptor, possibly including NS-specific desensitized states). Chang and colleagues have shown that orthosteric ligand affinity (muscimol or GABA) is greater in desensitized and activated (open) GABA$_A$Rs than in resting (closed) receptors, with estimated GABA K$_d$ values of 78.5 $\mu$M, 120 nM and 40 nM for the resting, activated and desensitized $\alpha_1\beta_2\gamma_2$ receptors respectively (*Chang et al., 2002*). Kinetic models also predict that a pre-active state should have higher affinity for an orthosteric agonist than the resting state (*Gielen and Corringer, 2018*). To distinguish between stabilization of a pre-active and a desensitized state, we co-applied $3\beta5\alpha$P with a high concentration of GABA. $3\beta5\alpha$P reduced the desensitization time constant but did not reduce peak current amplitude. While this result is consistent with stabilization of a desensitized rather than a pre-active state, it is ambiguous as it could also be explained by NAM-NS having a slower onset of effect than GABA because of slow access of the steroid to its binding site (*Li et al., 2007*). However, this result is supported by studies examining inhibitory postsynaptic currents (IPSCs) in which a NAM-NS can be pre-applied and the GABA concentration step is extremely rapid as well as by single channel studies. In cultured hippocampal neurons, the NAM-NS $3\beta5\beta$-THDOC significantly reduced IPSCs decay times but had no effect on IPSCs amplitude, consistent with a desensitizing effect (*Wang et al., 2002*). Single channel analyses provide the most definitive distinction between a pre-active and a desensitized state. Desensitization is predicted to shorten single channel clusters without affecting intracluster open or closed time distributions. In contrast, steroid-induced stabilization of a pre-active non-conducting state may be expected to lead to increased mean intracluster closed time. Single-channel studies examining the kinetic effects of the inhibitory steroids pregnenolone sulfate (*Akk et al., 2001*) or the $3\beta5\alpha$P analogue ($3\beta,5\alpha,17\beta$)−3-hydroxyandrostane-17-carbonitrile ($3\beta5\alpha$-ACN) (Akk, G.; unpublished data) have indeed observed reduced mean cluster duration with minimal changes in intracluster open and closed time properties, indicative of the steroids promoting receptor desensitization. The $\alpha_1$V256S mutation eliminated the PS-mediated reduction in cluster duration indicating that the mutation prevents PS-mediated desensitization (*Akk et al., 2001*). 3$\beta$-hydroxy steroids act similarly to PS, including exhibiting sensitivity to the $\alpha_1$V256S mutation (*Wang et al., 2002*). Overall, the preponderance of evidence indicates that 3$\beta$-NAM-NS such as $3\beta5\alpha$P inhibit GABA$_A$R currents by stabilizing a desensitized state.

Our experimental and modeling data demonstrate that NAM-NS such as $3\beta5\alpha$P or KK148 enhance orthosteric ligand binding by increasing the population of receptors in a desensitized state. It is, however, unclear if $3\beta5\alpha$P or KK148 can promote transition of resting receptors directly to a desensitized state, thus bypassing channel opening. We propose that in the presence of low

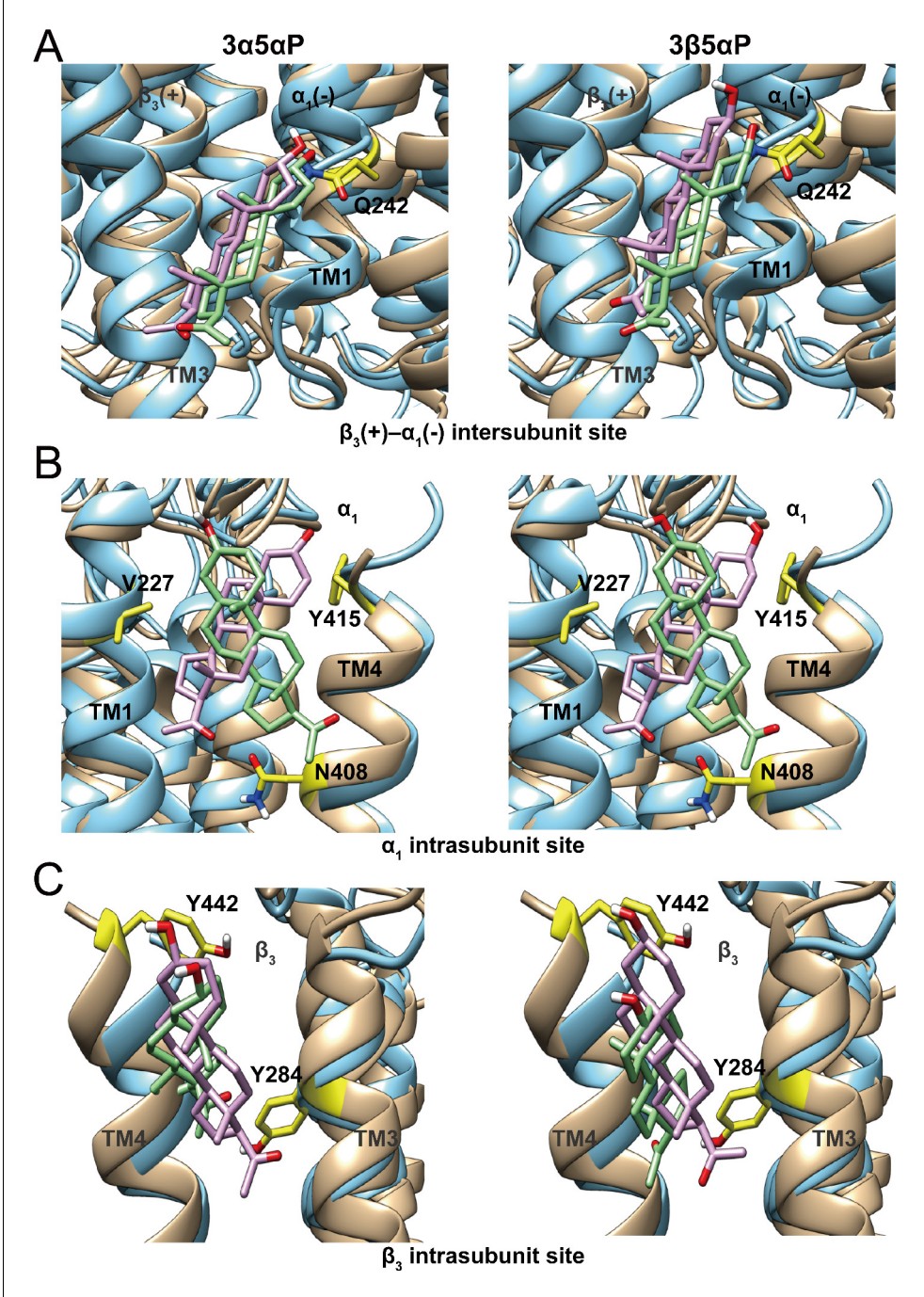

**Figure 11.** Comparison of allopregnanolone and epi-allopregnanolone docking poses within three neurosteroid binding pockets of the modeled $\alpha_1\beta_3$ GABA$_A$R TMD and the cryo-EM structure of an $\alpha_1\beta_3\gamma_2$ GABA$_A$R (PDB ID: 6I53). The two structures were read into UCSF Chimera and mutually aligned using MatchMaker. The $\alpha_1\beta_3$ model is shown in tan, while the $\alpha_1\beta_3\gamma_2$ structure is in cyan. (**A**) Representative poses for allopregnanolone (3α5αP) and epi-allopregnanolone (3β5αP) docked within the $\beta_3(+)$–$\alpha_1(-)$ intersubunit site, the poses for the $\alpha_1\beta_3$ model are in pink, while those for the $\alpha_1\beta_3\gamma_2$ structure are in light green. The $\alpha_1$Q242 side chain is shown in yellow. (**B**) Same as for (**A**) for the $\alpha_1$ intrasubunit site; also shown are the sidechains V227, Y415, and N408. (**C**) Same as (**A**) for the $\beta_3$ intrasubunit site; also shown are the sidechains Y284 and Y442. The Vina docking scores for 3α5αP and 3β5αP at each site in the $\alpha_1\beta_3$ model and the $\alpha_1\beta_3\gamma_2$ structure are shown in *Figure 11—source data 1*.

The online version of this article includes the following source data for figure 11:

**Source data 1.** Vina docking scores for allopregnanolone and epi-allopregnanolone at each site in $\alpha_1\beta_3$ model and $\alpha_1\beta_3\gamma_2$ GABA$_A$R structure.

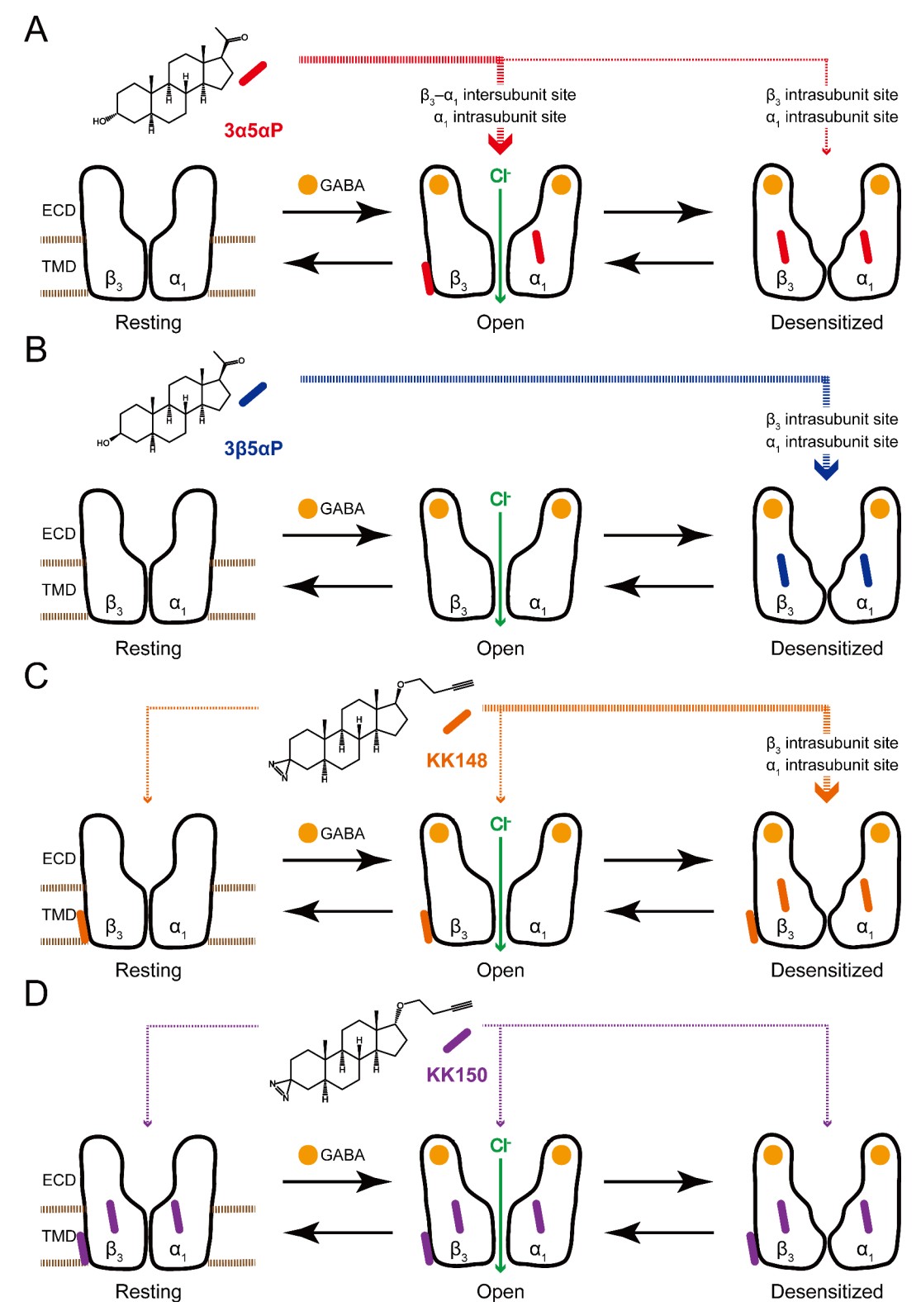

**Figure 12.** Neurosteroids preferentially stabilize GABA$_A$R in different states. (**A**) Model showing three fundamental conformational states that depict the channel function in the GABA$_A$R: a resting state; an open state; and a desensitized state. Agonist (GABA: ) binding shifts the equilibrium towards high-affinity states (open and desensitized). Allopregnanolone (3α5αP: ) allosterically stabilizes the high-affinity states (an open state through the β$_3$–α$_1$ intersubunit and the α$_1$ intrasubunit sites; a desensitized state through the β$_3$ intrasubunit site). The width of red arrows indicates relative affinities of

*Figure 12 continued on next page*

*Figure 12 continued*

3α5αP for the open or desensitized state of the receptor. (B) Same as (A) for epi-allopregnanolone (3β5αP: ). 3β5αP stabilizes a desensitized state through the β3 and α1 intrasubunit sites. (C) Same as (A) for KK148 ( ). KK148 allosterically stabilizes a desensitized state through the β3 and α1 intrasubunit sites, and equally stabilizes all three states of the receptor through the β3–α1 intersubunit site. The width of orange arrows indicates relative affinities of KK148 for each state of the receptor. (D) Same as (A) for KK150 ( ). KK150 equally stabilizes all three states of the receptor through the β3 and α1 intrasubunit sites, and the β3–α1 intersubunit site.

concentrations of orthosteric agonists (as in the [$^3$H]muscimol binding assays), there is a slow shift of receptors from resting through activated to a desensitized state with minimal change in the population of receptors in the activated state. The slow time course of accumulation of desensitized receptors is illustrated by experiments in which 10 µM 3β5αP is added to membranes that have been fully equilibrated with a low concentration (3 nM) of [$^3$H]muscimol and binding is measured as a function of time. Enhancement of [$^3$H]muscimol binding by 10 µM 3β5αP is slow, with a time constant of 4 min at 4˚C (τ = 3.97 ± 0.15 min: mean ± SEM, *n* = 4, *Figure 6—figure supplement 1*). In contrast, when α1β3 GABA$_A$Rs are exposed to long pulses of a high concentration of GABA, KK148- and 3β5αP-induced desensitization is rapid (*Figure 2A and C*), since in these conditions almost all of the receptors are either in an open or desensitized conformation and desensitization is not slowed by the transition from resting to open state (*Jones and Westbrook, 1995*). Thus, the slow enhancement of [$^3$H]muscimol binding by 3β5αP (*Figure 6—figure supplement 1*) is likely rate-limited by the transition of receptors to activated then to desensitized states at 3 nM muscimol rather than by 3β5αP binding. These time course experiments are most consistent with a model in which receptors preferentially progress from the resting to active to desensitized states, which are then stabilized by the NAM-NS.

The selective binding of 3β5αP to a subset of identified NS-binding sites provides an explanation for its NAM activity. 3β5αP stabilizes desensitized receptors by binding to the α1 and β3 intrasubunit sites, but does not activate the receptor because it does not bind to the intersubunit site. This site-selective binding is unexpected for several reasons. First, docking and free energy perturbation calculations in a prior study predicted that 3β5βP binds to the intersubunit site in a similar orientation and with free energies of binding that are equivalent to pregnanolone (3α5βP) (*Miller et al., 2017*). The modeling suggested that 3β5βP does not form a hydrogen bond with αQ242, a possible explanation for its lack of efficacy (*Miller et al., 2017*). Our docking studies also show similar binding energies and orientations of 3β5αP and 3α5αP binding in the β3(+)–α1(-) intersubunit site of either a homology model of the α1β3 receptor or the cryo-EM structure (PDB ID: 6I53) of the α1β3γ2 receptor in a lipid nanodisc (*Laverty et al., 2019*; *Figure 11*). We have also shown that binding affinity or docking scores of NS binding to the intersubunit site are not significantly affected by mutations (α1Q242L, α1Q242W, α1W246L) that eliminate NS activation, although binding orientation is altered (*Sugasawa et al., 2019*). These data indicate that NS binding in the intersubunit site is tolerant to significant changes in critical residues and NS ligand structure, and are consistent with our findings that NS analogues, such as KK148 and KK150, can bind to the intersubunit site but have no effect on activation (*Figures 1* and *4*). Thus, the peculiar lack of 3β5αP binding to the intersubunit site suggests that either: (1) details in the structure of the intersubunit site in the open conformation that explain the absence of 3β5αP binding are not apparent in current high-resolution structures or; (2) 3β5αP does not bind for other reasons. One plausible explanation is that 3β5αP, like cholesterol, has low chemical activity in the membrane and does not achieve sufficiently available concentration to bind in this site (*Lange and Steck, 2016*). This explanation would require that the chemical activity of 3β5αP differs between the inner and outer leaflets of a plasma membrane (presumably due to membrane lipid asymmetry) (*van Meer et al., 2008*; *Lorent et al., 2020*), since 3β5αP binds to the intrasubunit sites.

The functional analysis of mutations in each of the three NS-binding sites demonstrates that the activating and desensitizing effects of NS result from occupancy of distinct sites. In particular, binding of certain NS (3β5αP, KK148) to α1 and β3 intrasubunit sites modulates the open-desensitized equilibrium. Interestingly, lipid binding to intrasubunit pockets in bacterial pLGICs analogous to the α1 and β3 intrasubunit sites in GABA$_A$R, also modulates receptor desensitization; docosahexaenoic acid binding to an intrasubunit site in GLIC (*Basak et al., 2017*) and phosphatidylglycerol in ELIC (*Tong et al., 2019*) increase and decrease agonist-induced desensitization, respectively. The

combined results of mutational analyses and binding data demonstrate that the effects of various NS analogues are also a consequence of their efficacy at each of the sites they occupy. For example, KK148 and KK150 occupy the intersubunit site (*Figure 4C*), but do not activate GABA$_A$R currents (*Figure 1C–D*), and KK150 occupies both intrasubunit sites (*Figure 4B* and *Figure 4—figure supplement 1*) but does not desensitize the receptor (*Figure 2B*).

To explain the effects of the 3-substituted NS analogues, we propose a model in which NS-selective binding at three distinct binding sites on the GABA$_A$R preferentially stabilizes specific states (resting, open, desensitized) of the receptor (*Figure 12*). Orthosteric agonist (GABA or muscimol) binding shifts the equilibrium towards high-affinity states (open and desensitized). 3α5αP allosterically stabilizes the open state through binding to the β$_3$–α$_1$ intersubunit and α$_1$ intrasubunit sites and stabilizes the desensitized state through the β$_3$ intrasubunit site (*Figure 12A*). In contrast, 3β5αP preferentially stabilizes the desensitized state through binding to both intrasubunit sites (*Figure 12B*). KK148, like 3β5αP, stabilizes the desensitized state by binding to the intrasubunit sites (*Figure 12C*). KK148 also binds to the intersubunit site, presumably with no state-dependence, since it is neither an agonist nor an inverse-agonist (*Figure 1C* and *Figure 12C*). KK150, which neither activates nor desensitizes GABA$_A$Rs and is not an inverse agonist, binds to all three sites, again presumably with no-state dependence (*Figure 1D* and *Figure 12D*). This model predicts that KK148 should act as a competitive antagonist to PAM-NS at the intersubunit site. This model also predicts that KK150 should be a competitive NS antagonist at all three binding sites. Consistent with this prediction, KK150 antagonizes 3α5αP and KK148 enhancement of [$^3$H]muscimol binding (*Figure 8*).

The site-specific model of NS action (*Figure 12*) has significant implications for the synaptic mechanisms of PAM-NS action. At a synapse, GABA$_A$Rs are transiently exposed to high (mM) concentrations of GABA leading to a channel P$_{open}$ approaching one (*Farrant and Nusser, 2005*; *Feng and Forman, 2018*). GABA is quickly cleared from the synapse leading to rapid deactivation with minimal desensitization (*Jones and Westbrook, 1995*; *Overstreet et al., 2000*). In the presence of a PAM-NS, deactivation is slowed, resulting in a prolongation of the IPSC and increased inhibitory current (*Harrison et al., 1987a*; *Zhu and Vicini, 1997*; *Harrison et al., 1987b*; *Chakrabarti et al., 2016*). This effect is largely attributable to stabilization of the open state, presumably by binding to the intersubunit and α$_1$ intrasubunit binding sites. A second effect has been observed in which the PAM-NS 3α5αP (*Haage and Johansson, 1999*) and 3α5α-THDOC (*Zhu and Vicini, 1997*) prolong the slow component of GABA$_A$R desensitization and slow recovery from desensitization. This results in increased late channel openings (*Zhu and Vicini, 1997*; *Jones and Westbrook, 1995*) and IPSC prolongation (*Harrison et al., 1987b*; *Chakrabarti et al., 2016*). When the frequency of synaptic firing is rapid, the desensitizing effect of NS may also contribute to frequency-dependent reduction in IPSC amplitude (*Zhu and Vicini, 1997*; *Jones and Westbrook, 1996*). The desensitizing effect of 3α5αP is predominantly mediated by binding at the β$_3$ intrasubunit site. The balance between stabilization of the open and desensitized channels should be determined by the relative occupancies for the intersubunit site of the active receptor and the β$_3$ intrasubunit site of the desensitized receptor. Computational docking of 3α5αP to these sites indicates modest differences in affinity between the sites with a rank order affinity of: intersubunit > α$_1$ intrasubunit > β$_3$ intrasubunit sites (*Figure 11—source data 1*; *Chen et al., 2019*). Mutational analysis of the effects of NS on enhancement of [$^3$H]muscimol binding also indicates that 3α5αP has a lower affinity to the β$_3$ intrasubunit site (*Figure 6B*, *Table 1*). Thus ,binding to the β$_3$ intrasubunit site may serve as a negative feedback mechanism preventing excessive PAM-NS effects on synaptic currents.

We have identified specific NS-binding sites on α$_1$ and β$_3$ GABA$_A$Rs using photoaffinity labeling, but the structural details of NS interactions with these sites have not yet been elucidated. While several high-resolution structures of αβγ GABA$_A$Rs in a lipidic environment (nano-discs) with bound orthosteric or allosteric ligands have been published (*Laverty et al., 2019*; *Masiulis et al., 2019*; *Kim et al., 2020*), there are no available structures of a heteropentameric GABA$_A$R with a bound NS. There are, however, three X-ray structures of homopentameric, chimeric GABA$_A$Rs crystallized from detergent, all showing NS bound only in the intersubunit site (*Miller et al., 2017*; *Laverty et al., 2017*; *Chen et al., 2018*). Since we identified intrasubunit NS-binding sites by photolabeling full-length heteropentameric GABA$_A$Rs in native membranes, it is possible that either the non-natural ECD-TMD junctions or the detergent environment explain why neurosteroid binding to an intrasubunit site was not observed. However, we have also identified an α$_1$ intrasubunit NS

binding site using a 3α5αP analogue photolabeling reagent in a detergent-solubilized ELIC-$\alpha_1$ chimeric receptor (*Sugasawa et al., 2019*), indicating that NS can bind to this site in a detergent-solubilized chimeric receptor. It is important to note that there are many reasons why a bound NS (or any other ligand) may not be observed in X-ray or cryo-EM structures and that while these methods provide exquisite detail regarding the structure of ligand binding sites, they should not be regarded as a litmus test for the identification of ligand-binding sites. These reasons include: (1) a protein with steroid bound in the intrasubunit site may not form good crystals or may aggregate, precluding single particle analysis. (2) The conformation(s) observed in the X-ray or cryo-EM structure may not be the conformation to which NS preferentially binds in the intrasubunit site. (3) The ECD-TMD junction is the area of the protein that undergoes the most movement with activation and can be the least well-resolved portion of the transmembrane domains. (4) NS have multiple binding orientations and may be more mobile within the intrasubunit site making them more difficult to resolve. It is also important to consider that our analyses of the functional effects of NS binding were performed using $\alpha_1\beta_3$ GABA$_A$Rs. While the actions of inhibitory and potentiating NS in $\alpha\beta\gamma$ or $\alpha\beta\delta$ receptors are qualitatively similar to those observed in $\alpha\beta$ receptors, the trimeric receptors may have one less intrasubunit NS-binding site per pentamer, or an intrasubunit with different NS specificity or effect. Additionally, the presence of a $\gamma$-subunit has been shown to alter the conformational symmetry of the GABA$_A$R (*Laverty et al., 2019*; *Kim et al., 2020*; *Zhu et al., 2018*) and may influence NS binding to an intrasubunit site or its functional effects.

In summary, this study describes a unique NS pharmacology in which different NS analogues selectively bind to subsets of three sites on the $\alpha_1\beta_3$ GABA$_A$R, with each analogue exhibiting state-dependent binding at a given site. The combination of site-selectivity and state-dependence of binding determines whether a NS analogue is a PAM, a NAM or an antagonist of NS action at the GABA$_A$R. It seems likely that other GABA$_A$R subunit isoforms and heteropentameric subunit combinations will reveal additional NS-binding sites with distinct affinity and state-dependence for various analogues. The identification of potent agonists and antagonists for each of these sites will provide tools for understanding the biological effects of endogenous neurosteroids and potentially for the development of precision neurosteroid therapeutics.

## Materials and methods

### Construct design

The human $\alpha_1$ and $\beta_3$ GABA$_A$R subunits were subcloned into pcDNA3 for molecular manipulations and cRNA synthesis. Using QuikChange Site-Directed Mutagenesis Kit (Agilent Technologies, Santa Clara, CA), a FLAG tag was first added to the $\alpha_1$ subunit then an 8xHis tag was added to generate the following His-FLAG tag tandem (QPSLHHHHHHHHDYKDDDDKDEL), inserted between the fourth and fifth residues of the mature peptide. The $\alpha_1$ and $\beta_3$ subunits were then transferred into the pcDNA4/TO and pcDNA5/TO vectors (Thermo Fisher Scientific), respectively, for tetracycline inducible expression. Transient expression was done using the GABA$_A$R subunits rat $\alpha_1$FLAG (*Ueno et al., 1996*) and human $\beta_3$ obtained from Geoffrey White (Neurogen, Branford, CT), each were subcloned into pcDNA3 for molecular manipulations and cRNA synthesis. Point mutations were generated using the QuikChange Site-Directed Mutagenesis Kit (Agilent) and the coding region was fully sequenced prior to use. The cDNAs were linearized with XbaI (NEB Labs, Ipswich, MA), and the cRNAs were generated using T7 mMessage mMachine (Ambion, Austin, TX).

### Cell lines

Cell culture was performed as described in previous reports (*Chen et al., 2019*). The tetracycline inducible cell line T-REx-HEK293 (Thermo Fisher Scientific) was cultured under the following conditions: cells were maintained in DMEM/F-12 50/50 medium containing 10% fetal bovine serum (tetracycline-free, Takara, Mountain View, CA), penicillin (100 units/ml), streptomycin (100 g/ml), and blasticidin (2 μg/ml) at 37°C in a humidified atmosphere containing 5% $CO_2$. Cells were passaged twice each week, maintaining subconfluent cultures. Stably transfected cells were cultured as above with the addition of hygromycin (50 μg/ml) and zeocin (20 μg/ml). A stable cell line was generated by transfecting T-REx-HEK293 cells with human $\alpha_1$-8xHis-FLAG pcDNA4/TO and human $\beta_3$ pcDNA5/TO, in a 150 mm culture dish, using the Effectene transfection reagent (Qiagen, Germantown, MD).

Two days after transfection, selection of stably transfected cells was performed with hygromycin and zeocin until distinct colonies appeared. Medium was exchanged several times each week to maintain antibiotic selection. Individual clones were selected from the dish and transferred to 24-well plates for expansion of each clone selected. When the cells grew sufficiently, about 50% confluence, they were split into two other plates, one for a surface ELISA against the FLAG epitope and a second for protein assay, to normalize surface expression to cell number. The best eight clones were selected for expansion into 150 mm dishes, followed by [³H]muscimol binding to examine the receptor density. Once the best expressing clone was determined, the highest expressing cells of that clone were selected through fluorescence activated cell sorting. Transient transfections were done in HEK293S GnTI⁻ cells obtained from ATCC (CRL-3022) using Effectene (Qiagen). The identity of the cell lines has been authenticated using short tandem repeat analysis. Mycoplasma test performed on the cells used for these experiments was negative.

## Membrane protein preparation

Stably transfected cells were plated into dishes. After reaching 50% confluence, GABA$_A$ receptors were expressed by inducing cells with 1 µg/ml of doxycycline with the addition of 5 mM sodium butyrate. Cells were harvested 48 to 72 hr after induction. HEK cells, after induction, were grown to 100% confluence, harvested and washed with 10 mM potassium phosphate, 100 mM potassium chloride (pH 7.5) plus protease inhibitors (Sigma-Aldrich, St. Louis, MO) two times. The cells were collected by centrifugation at 1,000 g at 4°C for 5 min. The cells were homogenized with a glass mortar and a Teflon pestle for ten strokes on ice. The pellet containing the membrane proteins was collected after centrifugation at 20,000 g at 4°C for 45 min and resuspended in a buffer containing 10 mM potassium phosphate, 100 mM potassium chloride (pH 7.5). The protein concentration was determined with micro-BCA protein assay and membranes were stored at −80°C.

## Photolabeling and purification of α₁β₃ GABA$_A$R

The syntheses of neurosteroid photolabeling reagents (KK148, KK150, KK200, KK123) are detailed in previous reports (*Jiang et al., 2016*; *Cheng et al., 2018*). For all the photolabeling experiments, 10–20 mg of HEK cell membrane proteins (about 300 pmol [³H]muscimol binding) were thawed and resuspended in buffer containing 10 mM potassium phosphate, 100 mM potassium chloride (pH 7.5) and 1 mM GABA at a final concentration of 1.25 mg/ml. For the photolabeling competition experiments, 3 µM KK200 or KK123 in the presence of 30 µM competitor (3α5αP, KK148, KK150, and 3β5αP) or the same volume of ethanol was added to the membrane proteins and incubated on ice for 1 hr. The samples were then irradiated in a quartz cuvette for 5 min, by using a photoreactor emitting light at >320 nm. The membrane proteins were then collected by centrifugation at 20,000 g at 4°C for 45 min. The photolabeled membrane proteins were resuspended in lysis buffer containing 1% *n*-dodecyl-β-D-maltoside (DDM), 0.25% cholesteryl hemisuccinate (CHS), 50 mM Tris (pH 7.5), 150 mM NaCl, 2 mM CaCl₂, 5 mM KCl, 5 mM MgCl₂, 1 mM EDTA, 10% glycerol at a final concentration of 1 mg/ml. The membrane protein suspension was homogenized using a glass mortar and a Teflon pestle and incubated at 4°C overnight. The protein lysate was centrifuged at 20,000 g at 4°C for 45 min and supernatant was incubated with 0.5 ml anti-FLAG agarose (Sigma) at 4°C for 2 hr. The anti-FLAG agarose was then transferred to an empty column, followed by washing with 20 ml washing buffer (50 mM triethylammonium bicarbonate and 0.05% DDM). The GABA$_A$Rs were eluted with aliquots of 200 µg/ml FLAG tag peptide and 100 µg/ml 3X FLAG (ApexBio) in the washing buffer. The pooled eluates (9 ml) containing GABA$_A$Rs were concentrated to 100 µl using 100 kDa cut-off centrifugal filters.

## Tryptic middle-down MS analysis

The purified α₁β₃ GABA$_A$R (100 µl) was reduced with 5 mM tris(2-carboxyethyl)phosphine for 1 hr, alkylated with 5 mM *N*-ethylmaleimide (NEM) for 1 hr, and quenched with 5 mM dithiothreitol (DTT) for 15 min. These three steps were done at RT. Samples were then digested with 8 µg of trypsin for 7 days at 4°C to obtain maximal recovery of TMD peptides. The digestions were terminated by adding formic acid in a final concentration of 1%, followed directly by LC-MS analysis on an Orbitrap Elite mass spectrometer. 20 µl samples were injected into a home-packed PLRP-S (Agilent, Santa Clara, CA) column (10 cm ×75 µm, 300 Å), separated with a 145 min gradient from 10% to 90%

acetonitrile, and introduced to the mass spectrometer at 800 nl/min with a nanospray source. MS acquisition was set as a MS1 Orbitrap scan (resolution of 60,000) followed by top 20 MS2 Orbitrap scans (resolution of 15,000) using data-dependent acquisition, and exclusion of singly charged precursors. Fragmentation was performed using high-energy dissociation with normalized energy of 35%. Analysis of data sets was performed using Xcalibur (Thermo Fisher Scientific) to manually search for TM1, TM2, TM3 or TM4 tryptic peptides with or without neurosteroid photolabeling modifications. Photolabeling efficiency was estimated by generating extracted chromatograms of unlabeled and labeled peptides, determining the area under the curve, and calculating the abundance of labeled peptide/(unlabeled + labeled peptide). Analysis of statistical significance comparing the photolabeling efficiency of KK200 and KK123 for $\alpha_1\beta_3$ GABA$_A$R was determined using one-way ANOVA with Bonferroni's multiple comparisons test and paired $t$-test, respectively (Prism 6, GraphPad Software, San Diego, CA). MS2 spectra of photolabeled TMD peptides were analyzed by manual assignment of fragment ions with and without photolabeling modification. Fragment ions were accepted based on the presence of a monoisotopic mass within 20 ppm mass accuracy. In addition to manual analysis, PEAKS (Bioinformatics Solutions Inc, Waterloo, ON, Canada) database searches were performed for data sets of photolabeled $\alpha_1\beta_3$ GABA$_A$R. Search parameters were set for a precursor mass accuracy of 20 ppm, fragment ion accuracy of 0.1 Da, up to three missed cleavages on either end of the peptide, false discovery rate of 0.1%, and variable modifications of methionine oxidation, cysteine alkylation with NEM and DTT, and NS analogue photolabeling reagents on any amino acid.

## Radioligand-binding assays

[3H]muscimol-binding assays were performed using a previously described method (*Chen et al., 2019*). HEK cell membrane proteins (100 μg/ml final concentration) were incubated with 3 nM [3H] muscimol (30 Ci/mmol; PerkinElmer Life Sciences), neurosteroid (3 nM–30 μM) or etomidate (30 nM–200 μM) in different concentrations and binding buffer (10 mM potassium phosphate, 100 mM potassium chloride, pH 7.5) in a total volume of 1 ml. Assay tubes were incubated for 1 hr on ice in the dark. Nonspecific binding was determined by binding in the presence of 1 mM GABA. Membranes were collected on Whatman/GF-C glass filter papers using a Brandel cell harvester (Gaithersburg, MD). To perform [3H]muscimol binding isotherms, 100 μg/ml aliquots of membrane protein were incubated with 0.3 nM–1 μM [3H]muscimol for 1 hr on ice in the dark. The specific activity for [3H]muscimol concentrations from 30 nM to 1 μM was reduced to 2 Ci/mmol by dilution with nonradioactive muscimol. The membranes were collected on Whatman/GF-B glass filter papers using a vacuum manifold. Raw concentration-dependent total and nonspecific binding and calculated specific binding data from a representative experiment (WT receptors, no NS) are shown in *Figure 7— figure supplement 1*. For [3H]muscimol-binding experiments examining competitive interactions between neurosteroids, the combined neurosteroids (0.3 μM 3α5αP or 3 μM KK148 ±30 μM KK150) or the same volume of dimethyl sulfoxide (DMSO) were added to the membranes which were then incubated with 3 nM [3H]muscimol on ice for 1 hr. Time courses of neurosteroid [3H]muscimol binding enhancement were examined by adding 10 μM of neurosteroids (3α5αP, 3β5αP) to membranes that had been fully equilibrated with 3 nM [3H]muscimol for 1 hr on ice; binding was then measured as a function of time at 1, 3, 10, 30, 60 min. The membranes were collected on Whatman/GF-B glass filter papers using a vacuum manifold. Radioactivity bound to the filters was measured by liquid scintillation spectrometry using Bio-Safe II (Research Products International, Mount Prospect, IL).

## Radioligand binding to intact cells

HEK cells were harvested by gently washing dishes with buffer containing 10 mM sodium phosphate (pH 7.5), 150 mM sodium chloride twice. The cells were collected by centrifugation at 500 g at 4°C for 5 min, and resuspended in isotonic (10 mM sodium phosphate, 150 mM sodium chloride, pH 7.5) or hypotonic (10 mM sodium phosphate, pH 7.5) buffer to prepare two different conditions for radioligand binding to intact cells [isotonic buffer for cell surface receptors; hypotonic buffer for total receptors (cell surface receptors + intracellular receptors)]. The cells were incubated on ice for 2 hr, after which the sodium chloride concentration was adjusted to be isotonic before the radioligand binding procedure. HEK cells were aliquoted to assay tubes (20 samples/150 mm dish) in a total volume of 1 ml, and incubated with 3 nM [3H]muscimol ±10 μM KK148 for 1 hr on ice in the dark.

Nonspecific binding was determined by binding in the presence of 1 mM GABA. The membranes were collected on Whatman/GF-B glass filter papers using a vacuum manifold. Radioactivity bound to the filters was measured by liquid scintillation spectrometry using Bio-Safe II.

### Receptor expression in *Xenopus laevis* oocytes and electrophysiological recordings

The wild-type and mutant $\alpha_1\beta_3$ GABA$_A$R were expressed in oocytes from the African clawed frog (*Xenopus laevis*). Frogs were purchased from Xenopus 1 (Dexter, MI), and housed and cared for in a Washington University Animal Care Facility under the supervision of the Washington University Division of Comparative Medicine. Harvesting of oocytes was conducted under the Guide for the Care and Use of Laboratory Animals as adopted and promulgated by the National Institutes of Health. The animal protocol (No. 20180191) was approved by the Animal Studies Committee of Washington University in St. Louis. The oocytes were injected with a total of 12 ng cRNA. The ratio of cRNAs was 5:1 ratio ($\alpha_1$:$\beta_3$) to minimize the expression of $\beta_3$ homomeric receptors. Following injection, the oocytes were incubated in ND96 (96 mM NaCl, 2 mM KCl, 1.8 mM CaCl$_2$, 1 mM MgCl$_2$, 5 mM HEPES; pH 7.4) with supplements (2.5 mM Na pyruvate, 100 U/ml penicillin, 100 µg/ml streptomycin and 50 µg/ml gentamycin) at 16˚C for 2 days prior to conducting electrophysiological recordings. The electrophysiological recordings were conducted at room temperature using standard two-electrode voltage clamp. The oocytes were clamped at −60 mV. The chamber (RC-1Z, Warner Instruments, Hamden, CT) was perfused with ND96 at 5–8 ml/min. Solutions were gravity-applied from 30 ml glass syringes with glass luer slips via Teflon tubing. The current responses were amplified with an OC-725C amplifier (Warner Instruments, Hamden, CT), digitized with a Digidata 1200 series digitizer (Molecular Devices), and stored using pClamp (Molecular Devices). Current traces were analyzed with Clampfit (Molecular Devices). Activation by steroids (*Figure 1*) was tested by coapplying a steroid with 0.3 µM GABA (P$_{open}$ = 0.05–0.1). The desensitizing effects of steroids (*Figures 9–10*) were tested by coapplying a steroid with 1 mM (saturating) GABA, alone or in the presence of PB, during the steady-state phase of the current response. The combination of GABA and PB was used to activate some combinations of mutations to maintain a consistent, high peak open probability (0.55–0.95). In control experiments in WT $\alpha_1\beta_3$ GABA$_A$Rs, pentobarbital had no effect on 3$\beta$5$\alpha$P inhibition of steady-state current (data not shown). The stock solution of GABA was made in ND96 at 500 mM, stored in aliquots at −20˚C, and diluted on the day of experiment. The stock solution of muscimol was made at 20 mM in ND96 and stored at 4˚C. The steroids were dissolved in DMSO at 10–20 mM and stored at room temperature.

### Electrophysiological data analysis and simulations

The raw amplitudes of the current traces were converted to units of open probability through comparison to the peak response to 1 mM GABA + 50 µM propofol, that was considered to have a peak P$_{open}$ indistinguishable from 1 (*Chen et al., 2019*). The level of constitutive activity in the absence of any applied agonist was considered negligible and not included in this calculation. The converted current traces were analyzed in the framework of the three-state Resting-Open-Desensitized activation model (*Germann et al., 2019a*; *Germann et al., 2019b*). The model enables analysis and prediction of peak responses using four parameters that characterize the extent of constitutive activity (termed L; L = Resting/Open), affinity of the resting receptor to agonist (K$_C$), affinity of the open receptor to agonist (K$_O$), and the number of agonist binding sites (N). Analysis and prediction of steady-state responses requires an additional parameter, termed Q, that describes the equilibrium between open and desensitized receptors (Q = Open/Desensitized).

The P$_{open}$ of the peak response is expressed as:

$$P_{open,peak} = \frac{1}{1 + L\Gamma}$$

and the P$_{open}$ of the steady-state response as:

$$P_{open,steady-state} = \frac{1}{1 + \frac{1}{Q} + L\Gamma}$$

where

$$\Gamma = \left[\frac{(1+[X]/K_C)}{(1+[X]/K_O)}\right]^N$$

[X] is the concentration of agonist present, and other terms are as described above. In practice, the value of $L\Gamma$ was calculated using the experimentally determined $P_{open}$ of the peak response, and then used as a fixed value in estimating Q from $P_{open,steady-state}$.

The $P_{desensitized}$ was calculated using:

$$P_{desensitized} = \frac{1}{1 + Q + QL\Gamma}$$

The effect of 3β5αP on 1 mM GABA-elicited steady-state current was expressed through a change in the value of Q. The modified Q (termed Q*) was then used to predict changes in $P_{open}$ and $P_{desensitized}$ at low [GABA]. Calculated probabilities (e.g. $P_{open}$, $P_{desentitized}$) are reported as mean ± SD.

## Docking simulations

A model of the $\alpha_1\beta_3$ GABA$_A$R was developed using the crystal structure of the human $\beta_3$ homopentamer (PDB ID: 4COF) as a structural template (*Miller and Aricescu, 2014*). In this structure, the cytoplasmic loop was replaced with the sequence SQPARAA (*Jansen et al., 2008*). The pentamer subunits were organized as A $\alpha_1$, B $\beta_3$, C $\alpha_1$, D $\beta_3$, E $\beta_3$. The $\alpha_1$ sequence was aligned to the $\beta_3$ sequence using the program MUSCLE (*Edgar, 2004*). The pentameric alignment was then used as input for the program Modeller (*Sali and Blundell, 1993*), using 4COF as the template; a total of 25 models were generated. The best model as evaluated by the DOPE score (*Shen and Sali, 2006*) was then submitted to the H++ server (http://biophysics.cs.vt.edu) to determine charges and optimize hydrogen bonding. The optimized structure was then submitted to the PPM server (https://opm.phar.umich.edu/ppm_server) for orientation into a lipid membrane. The correctly oriented receptor was then submitted to the CHARMM-GUI Membrane Builder server (http://www.charmm-gui.org) to build the fully solvated, membrane bound system oriented into a 1-palmitoyl-2-oleoyl-sn-glycero-3-phosphatidylcholine (POPC) bilayer. The system was fully solvated with 40715 TIP3 water molecules and ionic strength set to 0.15 M KCl. The NAMD input files produced by CHARMM-GUI (*Lee et al., 2016*) use a seven-step process of gradually loosening constraints in the simulation prior to production runs. A 100 ns molecular dynamics trajectory was then obtained using the CHARMM36 force field and NAMD (*Lee et al., 2016*). The resulting trajectory was then processed using the utility mdtraj (*McGibbon et al., 2015*), to extract a snapshot of the receptor at each nanosecond of time frame. These structures were then mutually aligned by fitting the alpha carbons, providing a set of 100 mutually aligned structures used for docking studies. The docking was performed using Auto-Dock Vina (*Trott and Olson, 2010*) on each of the 100 snapshots in order to capture receptor flexibility. 3α5αP and 3β5αP were prepared by converting the sdf file from PubChem into a PDB file using Open Babel (*O'Boyle et al., 2011*), and Gasteiger charges and free torsion angles were determined by AutoDock Tools. Docking grid boxes were built for the $\beta_3$–$\alpha_1$ intersubunit, and the $\alpha_1$ and $\beta_3$ intrasubunit sites with dimensions of 15 × 15 × 15 Ångströms encompassing each binding pocket. Docking was limited to an energy range of 3 kcal from the best docking pose and was limited to a total of 20 unique poses. The docking results for a given site could result in a maximum of 2000 unique poses (20 poses × 100 receptor structures); these were then clustered geometrically using the program DIVCF (*Meslamani et al., 2009*). The resulting clusters were ranked by Vina score and cluster size, and then visually analyzed. A comparison of proposed NS-binding sites between the modeled $\alpha_1\beta_3$ GABA$_A$R TMD and the experimentally determined $\alpha_1\beta_3\gamma_2$ cryo-EM structure PDB ID: 6I53 (*Laverty et al., 2019*) was performed. The two structures were read into UCSF Chimera and mutually aligned using MatchMaker (*Meng et al., 2006*). Using the same Vina docking configuration files discussed above, 3α5αP and 3β5αP were then docked into the respective sites of the $\alpha_1\beta_3\gamma_2$ cryo-EM structure. The results are shown in *Figure 11* and *Figure 11—source data 1*; there was very little difference in the results between the modeled $\alpha_1\beta_3$ and the $\alpha_1\beta_3\gamma_2$ cryo-EM structure.

## Acknowledgements

The authors thank Drs. Charles F Zorumski, Steven Mennerick and Joseph Henry Steinbach for insightful advice and valuable suggestions.

## Additional information

### Funding

| Funder | Grant reference number | Author |
|---|---|---|
| National Institutes of Health | 2R01GM108799-05 | Alex S Evers<br>Douglas F Covey |
| National Institutes of Health | 5K08GM126336-03 | Wayland WL Cheng |
| National Institutes of Health | 5R01GM108580-06 | Gustav Akk |
| Taylor Family Institute for Innovative Psychiatric Research | | Zi-Wei Chen<br>David E Reichert<br>Douglas F Covey<br>Gustav Akk<br>Alex S Evers |

The funders had no role in study design, data collection and interpretation, or the decision to submit the work for publication.

### Author contributions

Yusuke Sugasawa, Data curation, Formal analysis, Investigation, Visualization, Writing - original draft, Project administration; Wayland WL Cheng, Methodology, Writing - review and editing; John R Bracamontes, Kathiresan Krishnan, Investigation, Methodology; Zi-Wei Chen, Investigation, Methodology, Writing - review and editing; Lei Wang, Data curation; Allison L Germann, Spencer R Pierce, Thomas C Senneff, Data curation, Investigation; David E Reichert, Data curation, Formal analysis, Investigation, Methodology, Writing - review and editing; Douglas F Covey, Resources, Supervision, Funding acquisition, Validation, Investigation, Methodology, Writing - review and editing; Gustav Akk, Conceptualization, Data curation, Formal analysis, Supervision, Funding acquisition, Validation, Investigation, Methodology, Writing - review and editing; Alex S Evers, Conceptualization, Resources, Software, Supervision, Funding acquisition, Validation, Visualization, Methodology, Writing - original draft, Project administration

### Author ORCIDs

Yusuke Sugasawa (iD) https://orcid.org/0000-0003-1607-0460
Wayland WL Cheng (iD) http://orcid.org/0000-0002-9529-9820
Zi-Wei Chen (iD) http://orcid.org/0000-0001-8601-2210
Alex S Evers (iD) https://orcid.org/0000-0002-0342-0575

### Decision letter and Author response

Decision letter https://doi.org/10.7554/eLife.55331.sa1
Author response https://doi.org/10.7554/eLife.55331.sa2

## Additional files

### Supplementary files

• Transparent reporting form

### Data availability

All data generated or analysed during this study are included in the manuscript and supporting files.

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
