## [Decision Letter]

**Acceptance summary:**

Neurosteroids are endogenous modulators of GABA_A_R receptors (GABA_A_Rs) in the brain. Neurosteroid binding to GABA_A_Rs can enhance or inhibit receptor function. In this study, combining radioligand binding assays, photoaffinity labeling, and electrophysiological analyses, the authors reveal that the functional effect of a given neurosteroid is dependent upon which binding sites are targeted, with binding to an intersubunit site causing positive allosteric modulation, whilst occupancy of intrasubunit sites promoting desensitization and negative allosteric modulation. Given the physiological significance of neurosteroids, elucidating how structurally similar neurosteroids can act as positive, negative or null modulators is clearly important.

**Decision letter after peer review:**

Thank you for submitting your article "Site-specific effects of neurosteroids on GABA_A_ receptor activation and desensitization" for consideration by *eLife*. Your article has been reviewed by three peer reviewers, and the evaluation has been overseen by a Reviewing Editor and Richard Aldrich as the Senior Editor. The reviewers have opted to remain anonymous.

The reviewers have discussed the reviews with one another and the Reviewing Editor has drafted this decision to help you prepare a revised submission.

As the editors have judged that your manuscript is of interest, but as described below that additional experiments are required before it is published, we would like to draw your attention to changes in our revision policy that we have made in response to COVID-19 (https://elifesciences.org/articles/57162). First, because many researchers have temporarily lost access to the labs, we will give authors as much time as they need to submit revised manuscripts. We are also offering, if you choose, to post the manuscript to bioRxiv (if it is not already there) along with this decision letter and a formal designation that the manuscript is 'in revision at *eLife*'. Please let us know if you would like to pursue this option. (If your work is more suitable for medRxiv, you will need to post the preprint yourself, as the mechanisms for us to do so are still in development.)

Summary:

This report provides significant new information about the mechanisms of neurosteroid enhancement and inhibition of GABA_A_ receptor (GABA _A_R) function. This study builds on an earlier investigation by the same group (Chen et al., 2019) showing that photoactive NS ligands can bind to three distinct sites on α_1_β_3_ GABA _A_Rs – the canonical intersubunit site at the interface between the transmembrane domains (TMDs) of adjacent subunits and additional intrasubunit sites located within the TMDs of the α and β subunits. In the current study, combining [^3^H]muscimol radioligand binding assays, site identification by photoaffinity labeling, and electrophysiological analyses of steroid modulation of wildtype and mutant α_1_β_3_ GABA_A_Rs, the authors suggest that the overall functional effect of a given NS molecule is dependent upon which binding sites are targeted, with binding to the intersubunit site causing positive allosteric modulation (PAM), whilst occupancy of the intrasubunit sites appear to promote desensitization and negative allosteric modulation (NAM). Given the physiological significance of neurosteroids, elucidating how these structurally similar compounds can act as positive, negative or null modulators is clearly important.

Addressing the following essential revisions will strengthen the manuscript.

Essential Revisions:

1) The electrophysiological data presented (changes in steady state desensitization current magnitudes) is insufficient to conclude that NAM steroids inhibit GABA_A_R function by stabilizing a desensitized state. Additional experiments such as co-application of agonist + NS and monitoring desensitization kinetics would be informative. Measuring the rate of recovery from agonist-induced desensitization in the presence of neurosteroids might also be helpful. While the data presented can be interpreted as changes in desensitization, the authors should discuss that alternative models are also possible. For example, it has been proposed that selectively stabilizing a pre-active state can result in changes in macroscopic desensitization (Gielan and Corringer, 2018).

2) Mutant receptors were not assayed for their sensitivities to agonist before measuring effects of neurosteroids. The functional assays and binding experiments need to be done at a consistent fractional EC value for each mutant construct being analyzed. For example, if the apparent Kd for muscimol has shifted substantially, the observed potentiation of muscimol binding by a neurosteroid will be artificially high or low. The is also true for experiments measuring neurosteroid potentiation/inhibition of functional activation by GABA_A_.

3) In the Results section, there are concerns about quantitatively comparing electrophys data and [^3^H]muscimol data (measured at different agonist concentrations and time periods). Are the methods reliable enough to infer that the small changes in P_open_ and P_desensitized_ are real? In some cases, data are not shown. Inherent methodological limitations of two-electrode voltage clamping (e.g. slow ligand exchange) raises concerns that authors are over interpreting the data. As it stands, the comparison seems to be a bit of a reach and in this reviewers' opinion does not significantly add to the paper.

4) While having three distinct sites for NS binding to GABA_A_Rs does fit with aspects of the data, it's noteworthy that with the suggested model, there are three ligands that bind to all three sites, 3α5αP, KK148 and KK150, but each has a distinct functional profile, PAM, NAM via stabilizing desensitization, and competitive antagonist, respectively. This implies that divergence in function is dependent upon differential binding/efficacy at these three sites, presumably due to the ligand sitting in each site in a different orientation. While the observation from the [^3^H]muscimol binding experiments suggests that 3α5αP binds to the β_3_ intrasubunit site with lower affinity, the data presented in Figure 6B also suggest that binding of 3α5αP to the intersubunit and α_1_ intrasubunit sites works synergistically to increase muscimol binding. The reasoning being because with both sites intact, the Emax for muscimol binding is 374%, whereas mutating these sites individually causes similar decreases in Emax (to 159% and 146%). This implies an allosteric interaction between these binding sites, a conclusion which the authors also reach in their previous publication (Chen et al., 2019). This makes interpretation of the effects of mutations in these two sites (and possibly also the β intrasubunit site) difficult to interpret and to use to specifically dissociate a mutations effects on NS actions to binding to one particular site. The authors need to thoroughly discuss this concern/limitation.

5) The demonstration that steroids apparently enhance [^3^H]muscimol binding affinity without changing the number of sites (Figure 6—figure supplement 1) is in contrast to past reports from multiple labs that [^3^H]muscimol binding (to brain membranes) is characterized by high and low affinity components and that steroids and other GABA_A_R positive allosteric modulators increase the number of high affinity sites with little effect on their binding affinity. Please discuss. In addition, we would like to see presented in supplementary material representative experimentally determined [^3^H]muscimol binding curves (total and non-specific vs. [^3^H]musc concentration, not just the calculated Bspec of Figure 6—figure supplement 1). In their Materials and methods (subsection “Radioligand binding assays”) they say that they determined [^3^H]muscimol binding isotherms from 0.3 nM to 1 μm [^3^H]muscimol at a radiochemical specific activity of 2 Ci/mmol. It is surprising that they can go to μM concentrations with such small uncertainties, and it is crucial to their claim steroids produce only shifts of affinity, not shifts of B_max_.

6) [^3^H]muscimol binding is measured on cell homogenates over a time scale of hours. There seems no reason to "infer" that 3α5αβP increases [^3^H]muscimol binding by stabilizing an active state while 3β5αP stabilizes a desensitized state. By my reading, the previous studies (Akk et al., 2001; Wang et al., 2002) report that the αV256S mutation removes the "inhibitory" effects of sulfated steroids and 3β5αP, not the "desensitizing" effects, this should be more clearly articulated in this manuscript. This report will be strengthened by avoiding unnecessary overinterpretation, and leave it for future studies to determine whether there is any measurable quantity of receptors in an active state under the conditions of the [^3^H]muscimol equilibrium binding assay.

7) Given the expectations that some of the neurosteroids stabilize a desensitized state, do they "fit" in the proposed intrasubunit sites in, for example, one of the published presumed desensitized-state structures of the α_1_β_3_γ_2_ receptor?

8) A more thorough discussion of why recently solved GABA_A_R structures have not resolved intrasubunit neurosteroid binding sites is warranted.

---

## [Author Response]

Essential Revisions:1) The electrophysiological data presented (changes in steady state desensitization current magnitudes) is insufficient to conclude that NAM steroids inhibit GABA_A_R function by stabilizing a desensitized state. Additional experiments such as co-application of agonist + NS and monitoring desensitization kinetics would be informative. Measuring the rate of recovery from agonist-induced desensitization in the presence of neurosteroids might also be helpful. While the data presented can be interpreted as changes in desensitization, the authors should discuss that alternative models are also possible. For example, it has been proposed that selectively stabilizing a pre-active state can result in changes in macroscopic desensitization (Gielan and Corringer, 2018).

The high affinity, non-conducting state stabilized by NAM-neurosteroids could, in principle, be either a pre-active or a post-active (desensitized) state. Indeed, as Gielen and Corringer (Gielan and Corringer, 2018) emphasize in their highly-detailed and well-researched review article, selective stabilization of a pre-active state can account for faster rate of current decay and lower steady-state current. However, as also pointed out, stabilization of the pre-active state is expected to reduce the peak response to agonist, whereas stabilization of a [post-active] desensitized state is predicted to be without effect on peak response (Figure 10 in Gielen and Corringer, 2018). As requested, we have now included a figure (Figure 2—figure supplement 1) showing that co-application of saturating GABA (1 mM) with 3β5αP does not reduce peak current amplitude, but does increase the rate of current decay. While this result appears consistent with stabilization of a desensitized state rather than a pre-active state, it is ambiguous because it is possible that the steroid has a slower onset than GABA, thus minimizing the effect on peak current. However, prior studies have also shown that GABA_A_R NAM-NS, such as 3β5β-THDOC (Wang et al., 2002) or pregnenolone sulfate (PS) (Eisenman et al., 2003, PMID: 12938673) do not reduce the peak amplitude of autaptic IPSCs at concentrations where the effect on decay time is evident. Also, as shown by Akk, exposure to PS (Akk et al., 2001) or 3β5α-ACN (unpublished results; Author response image 1) reduces the mean single-channel cluster duration but does not affect the intracluster open and closed time distributions in steady-state single-channel recordings. This is indicative of NAM-NS-facilitated entry into a long-lived non-conducting state (although not necessarily the "true" desensitized state observed in the continuous presence of the transmitter) without modification of pre-active transitions. All of these points are now incorporated into modified Results and Discussion.

**Author response image 1. sa2fig1:** Single channel recordings from HEK-293 cells transfected with α_1_β_2_γ_2_ GABA_A_ receptors and exposed to 50 µM GABA ± 3β5α-ACN. The NAM-neurosteroid reduces cluster duration with minimal effects on intracluster closed time distribution.

2) Mutant receptors were not assayed for their sensitivities to agonist before measuring effects of neurosteroids. The functional assays and binding experiments need to be done at a consistent fractional EC value for each mutant construct being analyzed. For example, if the apparent Kd for muscimol has shifted substantially, the observed potentiation of muscimol binding by a neurosteroid will be artificially high or low. The is also true for experiments measuring neurosteroid potentiation/inhibition of functional activation by GABA.

We agree that mutations could alter orthosteric ligand affinity and that the effects of neurosteroids on [^3^H]muscimol binding should be measured at a consistent fractional EC value. To address this, we generated [^3^H]muscimol binding curves and determined muscimol K_d_ values for receptors with mutations in each of the neurosteroid binding sites. The binding curves are now shown in Figure 7B and the K_d_ and B_max_ values in Figure 7—source data 1. The mutations did not cause a significant change in muscimol K_d_ from wild-type. Therefore the concentration-dependent effects of neurosteroids on [^3^H]muscimol binding were measured at a fixed concentration (3 nM; ∼EC_5_) as in the original manuscript (Figure 6) (subsection “Orthosteric ligand binding enhancement by NS analogues is mediated by distinct sites”). The electrophysiological experiments were conducted at a relatively narrow range of peak P_open_ values (~0.55–0.95). This was achieved by activating the receptors with 1 mM GABA, alone or in the presence of 25–40 µM pentobarbital. This is now described in the Materials and methods and Results sections.

3) In the Results section, there are concerns about quantitatively comparing electrophys data and [^3^H]muscimol data (measured at different agonist concentrations and time periods). Are the methods reliable enough to infer that the small changes in P_open_ and P_desensitized_ are real? In some cases, data are not shown. Inherent methodological limitations of two-electrode voltage clamping (e.g. slow ligand exchange) raises concerns that authors are over interpreting the data. As it stands, the comparison seems to be a bit of a reach and in this reviewers' opinion does not significantly add to the paper.

We think it is important for the coherence of the manuscript and for the subsequent discussion that we explain how stabilization of a desensitized state can result in relatively small changes in steady-state current associated with relatively large changes in the occupancy of high-affinity states. We have simplified the data presentation, eliminating the data on low GABA concentrations and making a clearer summary (subsection “Quantitative comparison of the effects of 3β5αP on [^3^H]muscimol binding and receptor desensitization”). The mechanisms of enhancement of [^3^H]muscimol binding by allosteric activators and inhibitors are described in detail in our recent publication (Akk et al., 2020).

We do agree that the measured changes in current are small, but they are precise because each experiment served as its own control; a steady-state current was achieved during continuous agonist administration and the response to 3β5αP was then measured. This is illustrated by the sample trace showing the small change in P_open_ (0.011 to 0.009) elicited by 3β5αP following muscimol administration (Figure 2 in this letter). We also note that each day before experiments the drug application system was initially tested with a common activator (e.g., a low concentration of GABA or muscimol) to verify that equal-size responses were obtained with the drug applied from different syringes.

Figure 2: Effect of 3β5αP on steady state current selected by 20 nM muscimol. Mean change in steady-state P_open_ is from 0.011 to 0.009.

4) While having three distinct sites for NS binding to GABA_A_Rs does fit with aspects of the data, it's noteworthy that with the suggested model, there are three ligands that bind to all three sites, 3α5αP, KK148 and KK150, but each has a distinct functional profile, PAM, NAM via stabilizing desensitization, and competitive antagonist, respectively. This implies that divergence in function is dependent upon differential binding/efficacy at these three sites, presumably due to the ligand sitting in each site in a different orientation. While the observation from the [^3^H]muscimol binding experiments suggests that 3α5αP binds to the β_3_ intrasubunit site with lower affinity, the data presented in Figure 6B also suggest that binding of 3α5αP to the intersubunit and a_1_ intrasubunit sites works synergistically to increase muscimol binding. The reasoning being because with both sites intact, the Emax for muscimol binding is 374%, whereas mutating these sites individually causes similar decreases in Emax (to 159% and 146%). This implies an allosteric interaction between these binding sites, a conclusion which the authors also reach in their previous publication (Chen et al., 2019). This makes interpretation of the effects of mutations in these two sites (and possibly also the β intrasubunit site) difficult to interpret and to use to specifically dissociate a mutations effects on NS actions to binding to one particular site. The authors need to thoroughly discuss this concern/limitation.

We agree with the reviewer that the divergence in function between the 3 neurosteroid analogues that bind at all three sites is dependent on differential binding/efficacy at each site, which is presumably due to each ligand binding in each site with a different orientation. Moreover, each of the ligands must have preferential affinity/efficacy for specific sites in different conformations (R-A-D). This point is illustrated by the model in Figure 12.

The data showing that mutations in specific neurosteroid binding sites decrease the effect of the neurosteroid (either enhancement of [^3^H]muscimol binding or inhibition of steady-state current) indicate that neurosteroid binding at that site contributes to the effect. The % inhibition produced by the mutation does not denote the quantitative contribution of that site to the effect and the % effect of mutations in different sites should not be expected to be algebraically additive to the % effect in wild type receptors. In our previous publication (Chen et al., 2019) we proposed that super additivity of % inhibition of allopregnanolone-potentiated currents observed with mutations in the intersubunit and intrasubunit binding sites indicates an allosteric interaction between the two sites. But there are other explanations. The mutational data can be parsimoniously explained by neurosteroids binding and acting independently at each binding site with a non-linear summation of the effects (channel activation, enhanced muscimol binding or enhanced desensitization).

In its simplest form, our interpretation is based on the application of a two state (Resting-Activated) Monod-Wyman-Changeux concerted transition model to examine the effects of mutations on activation by agonist X. It is assumed that the receptor has two binding sites for X. MT1 (mutant #1) reduces activation through site 1 and MT2 reduces activation through site 2. The effect of each mutation is to independently change *c* (the ratio of the K_d_ of X in the active state to its K_d_ in the resting state) in one of the two binding sites. As an example, if either mutation increases c (reduces gating efficacy) in its site from 0.01 to 0.3 the following results are predicted: In wild-type receptor, P_o_,peak is 0.91 (simulation done at saturating concentration of X; L=1000). In MT1 (as well as in MT2), P_o_,peak is 0.25. And in the double mutant (MT1 + MT2), P_o_,peak is 0.01. So, either mutation individually elicits a 73% loss of gating, while the combination elicits a 99% loss of gating. Hence, the effects of independently acting mutations are not additive numerically. The same logic applies to the effects of mutations on other NS actions (enhancement of [^3^H]muscimol binding or inhibition of steady state current.) The independence of the intersubunit and α_1_ intrasubunit sites is also supported by the data in Figure 6C and D which show that the α_1_Q242L mutation has no effect on modulation of [^3^H]muscimol binding by KK148 or 3β5αP, which do and do not occupy the intersubunit site respectively. If the α_1_Q242L mutation itself or NS occupancy of the intersubunit site allosterically modulates the α_1_ intrasubunit site, there should be an effect on modulation of [^3^H]muscimol binding.

5) The demonstration that steroids apparently enhance [^3^H]muscimol binding affinity without changing the number of sites (Figure 6—figure supplement 1) is in contrast to past reports from multiple labs that [^3^H]muscimol binding (to brain membranes) is characterized by high and low affinity components and that steroids and other GABA_A_R positive allosteric modulators increase the number of high affinity sites with little effect on their binding affinity. Please discuss. In addition, we would like to see presented in supplementary material representative experimentally determined [^3^H]muscimol binding curves (total and non-specific vs. [^3^H]musc concentration, not just the calculated Bspec of Figure 6—figure supplement 1). In their Materials and methods (subsection “Radioligand binding assays”) they say that they determined [^3^H]muscimol binding isotherms from 0.3 nM to 1 μm [^3^H]muscimol at a radiochemical specific activity of 2 Ci/mmol. It is surprising that they can go to μM concentrations with such small uncertainties, and it is crucial to their claim steroids produce only shifts of affinity, not shifts of B_max_.

This is a very messy topic! There is a substantial literature from the 1990’s showing two-component binding curves for [^3^H]muscimol in brain with a high and low affinity component (e.g. Harrison and Simmonds, 1984). Whether this is based on heterogeneity of receptor subtypes, or multiple states of the GABA_A_R is unresolved. The reported effects of neurosteroids on these two-component systems is also variable, generally showing an increase in the B_max_ of the high affinity component. Whether this is due to recruitment of receptors or conversion of low affinity receptors to high affinity receptors is also unresolved. The question of whether neurosteroids reduce the K_d_ of the high affinity component is difficult to discern in these studies because they often resolve two poorly separated components from binding curves with shallow Hill slopes.

Our α_1_β_3_ expression system produced a single-component [^3^H]muscimol binding curve with a K_d_ (36 nM) that is higher than the high affinity K_d_ observed in brain (15 nM; Harrison and Simmonds, 1984). Our [^3^H]muscimol binding curve is extremely similar to the single component curve reported for [^3^H]muscimol binding to expressed α_1_β_3_γ_2_ receptors (Dostalova et al., 2014) with a reported K_d_ of 49 nM. The single component curve allowed us to observe a modest neurosteroid-induced left-shift in K_d_ to 15 nM that might be difficult to discern in a more complex 2-component system. We do not have a mechanistic explanation for why a two-component binding curve is observed in brain and we observed a single component curve. While this is a very interesting question, it does not seem germane to the current work and we have not discussed it in the manuscript.

Representative experimentally determined [^3^H]muscimol binding curves (raw concentration-dependent total and nonspecific binding and calculated specific binding data from a representative experiment in WT receptors) are now shown in Figure 7—figure supplement 1 (subsection “Radioligand binding assays”).

Thank you for pointing out the issue of [^3^H]muscimol specific activity. To clarify, in generating [^3^H]muscimol binding curves we used a specific activity of 30 Ci/mmol for muscimol concentrations of < 30 nM and a specific activity of 2 Ci/mmol for muscimol concentrations from 30 nM to 1 μM. This is now presented in the Materials and methods.

6) [^3^H]muscimol binding is measured on cell homogenates over a time scale of hours. There seems no reason to "infer" that 3α5αβP increases [^3^H]muscimol binding by stabilizing an active state while 3β5αP stabilizes a desensitized state. By my reading, the previous studies (13,14) report that the αV256S mutation removes the "inhibitory" effects of sulfated steroids and 3β5αP, not the "desensitizing" effects, this should be more clearly articulated in this manuscript. This report will be strengthened by avoiding unnecessary overinterpretation, and leave it for future studies to determine whether there is any measurable quantity of receptors in an active state under the conditions of the [^3^H]muscimol equilibrium binding assay.

While [^3^H]muscimol binding was measured over an hour, steroid modulation occurs over minutes (Figure 6—figure supplement 1) and both enhancement of [^3^H]muscimol binding and inhibition of steady-state current are measured at equilibrium. Since the α_1_V256S mutation does not abolish 3α5αP enhancement of [^3^H]muscimol binding it follows that the enhancement is not due to stabilization of a desensitized (see below) state of the receptor.

The reviewer raises the question of whether reduced neurosteroid inhibition in α_1_V256S mutant receptors equates to reduced desensitization. There is significant evidence that the inhibition produced by PS represents desensitization. Akk et al., 2001, showed that exposure to PS reduces the mean duration of single-channel cluster by entry into a long-lived non-conducting state (Figure 1 in Akk et al., 2001). This effect was essentially eliminated by the α_1_V256S mutation (Figure 6 in Akk et al., 2001). The effect was termed "block" due to absence of structural evidence for desensitization. In other words, it was, and still is, not possible to distinguish the PS-elicited non-conducting state from the transmitter-elicited non-conducting state (i.e., true desensitization). The intracluster open and closed times were not affected by PS, indicating that PS does not act by stabilizing the pre-active state (as discussed by Gielen and Corringer, 2018, for DHA inhibition of GLIC). The major conclusion by Wang et al., 2002, was that 3β-hydroxy steroids act similarly to PS, including exhibiting sensitivity to the α_1_V256S mutation.

7) Given the expectations that some of the neurosteroids stabilize a desensitized state, do they "fit" in the proposed intrasubunit sites in, for example, one of the published presumed desensitized-state structures of the α_1_β_3_γ_2_ receptor?

Docking of 3β5αP and 3α5αP with the cryo-EM structure of a α_1_β_3_γ_2_ receptor in a lipid nanodisc (6I53) and in a homology model of a α_1_β_3_ receptor are now shown in Figure 11. The neurosteroids dock in the intersubunit site and α_1_ and β_3_ intrasubunit sites. The preferred poses and docking scores are very similar between the α_1_β_3_ model and the α_1_β_3_γ_2_ structure (Materials and methods).

8) A more thorough discussion of why recently solved GABA_A_R structures have not resolved intrasubunit neurosteroid binding sites is warranted.

The only published structures of a GABA_A_ receptor with a neurosteroid are X-ray structures of chimeric GABA_A_ receptors, crystallized from detergent (Miller et al., 2017; Laverty et al., 2017; Chen et al., 2018). Our identification of the neurosteroid intrasubunit binding sites was done by photolabeling in native membranes. Either the non-natural ECD-TMD junctions or the binding in detergent are possible reasons that neurosteroid binding to an intrasubunit site was not observed. However, we did photolabel the α_1_ intrasubunit site with a 3α5αP analogue photolabeling reagent in a detergent-solubilized α_1_-ELIC chimeric receptor (Sugasawa et al., 2019), indicating that neurosteroids can bind to this site in a detergent-solubilized chimeric receptor. There are many reasons why a ligand may not be observed in X-ray or cryo-EM structures. These reasons include: 1) the protein with steroid bound in the intrasubunit site may not form good crystals or may aggregate, precluding single particle analysis. 2) The conformation observed in the X-ray or cryo-EM structure may not be the conformation to which neurosteroid preferentially binds in the intrasubunit site. 3) The ECD-TMD junction is the area of the protein that undergoes the most movement with activation and can be the least well-resolved portion of the transmembrane domains. 4) Neurosteroids have multiple binding orientations and may be more mobile within the intrasubunit site making them more difficult to resolve.